# Synthetic control of correlated disorder in UiO-66 frameworks

Sergio Tatay[1] ✉, Sonia Martínez-Giménez[1], Ana Rubio-Gaspar[1], Eloy Gómez-Oliveira [1], Javier Castells-Gil [1], Zhuoya Dong [2,3], Álvaro Mayoral [3], Neyvis Almora-Barrios[1], Natalia M. Padial[1] & Carlos Martí-Gastaldo [1] ✉

Changing the perception of defects as imperfections in crystalline frameworks into correlated domains amenable to chemical control and targeted design might offer opportunities for the design of porous materials with superior performance or distinctive behavior in catalysis, separation, storage, or guest recognition. From a chemical standpoint, the establishment of synthetic protocols adapted to control the generation and growth of correlated disorder is crucial to consider defect engineering a practicable route towards adjusting framework function. By using UiO-66 as experimental platform, we systematically explored the framework chemical space of the corresponding defective materials. Periodic disorder arising from controlled generation and growth of missing cluster vacancies can be chemically controlled by the relative concentration of linker and modulator, which has been used to isolate a crystallographically pure "disordered" *reo* phase. Cs-corrected scanning transmission electron microscopy is used to proof the coexistence of correlated domains of missing linker and cluster vacancies, whose relative sizes are fixed by the linker concentration. The relative distribution of correlated disorder in the porosity and catalytic activity of the material reveals that, contrarily to the common belief, surpassing a certain defect concentration threshold can have a detrimental effect.

Metal-Organic Frameworks (MOFs) are chemically tunable reticular platforms with tailorable composition, structure, and porosities that have encountered wide applications in fields as catalysis, gas storage/separation or biology[1,2]. Though these frameworks are conventionally defined as "ideal" in terms of the periodic and ordered distribution of metal nodes and organic linkers, their assembly often involves important deviations from the ideal stoichiometry even for subtle differences between synthetic protocols[3]. These compositional deviations converge in the general concept of defective MOFs[4-6], that has gained increasing importance in the last decade thanks to the experimental impact of these compositional defects in their

mechanical properties[6], or their performance in gas adsorption[7,8], separation[9,10] and catalysis[11-15]. In this context, the 12-connected (c) Zr-based UiO-66 MOF with *fcu* topology[16] is the canonical platform for defect engineering because of the capability of the $Zr_6$ cluster to accommodate a high number of linker vacancies without provoking a collapse of the framework. The presence of linker defects in Zr-UiO-66 was first observed thermogravimetrically in 2011[17], and later confirmed through high-resolution neutron[18], and X-ray single crystal-diffraction studies[19,20]. While it was initially assumed that defects existed in the form of randomly distributed missing linkers (ML), a turning point arrived in 2014 when weak and broad superlattice powder X-ray

[1]Instituto de Ciencia Molecular, Universitat de València, Paterna 46980, Spain. [2]School of Physical Science and Technology & Shanghai Key Laboratory of High-resolution Electron Microscopy, ShanghaiTech University, Shanghai 201210, P. R. China. [3]Instituto de Nanociencia y Materiales de Aragón (INMA), CSIC-Universidad de Zaragoza, Zaragoza 50009, Spain. ✉e-mail: sergio.tatay@uv.es; carlos.marti@uv.es

diffraction peaks in UiO-66[21–23], were unambiguously attributed to a secondary crystalline phase based on metal cluster vacancies having practically the same unit cell size but lower symmetry[24]. The symmetry lowering results from the removal of one cluster and all connected linkers in the 12-c *fcu* net to generate defective 8-c domains with a *reo* topology indicative of structurally correlated vacancies. Later, by using low-dose high-resolution transmission electron microscopy (TEM) and scanning electron diffraction (SED), these missing cluster (MC) nanodomains were directly visualized and found to co-exist with related ML variants[25,26]. Although the concept of defectivity is often understood as the introduction of punctual, aperiodic vacancies by random removal of the molecular components of the framework, these seminal reports anticipated how the disorder generated in these frameworks by changes in composition is not necessarily random and can be structurally correlated into periodic domains. The implications that controlling periodic disorder might have in adjusting crystal function are certainly fascinating[27–29], but arguably reliant on the establishment of synthetic routes adapted to systematic control of chemical composition for directing the generation and growth of correlated defect domains. Here, we report how the systematic experimental design and data analysis can be used to gain control over defect formation in Hf-UiO-66. Systematic variation of temperature, modulator, and linker concentrations can be used to isolate increasing relative percentages of defective domains up to near 90% crystallographically pure *reo* phases. The data extracted from X-ray diffraction Rietveld refinement and TEM reveal an intimate interplay between ML and MC vacancies, that operate in parallel, and cannot be disentangled in the formation of correlated disorder. We also demonstrate that the positive effect of linker vacancies in the formation of bigger correlated domains, also has a detrimental impact on the structural stability of these samples, which is here correlated with drastic changes in their catalytic activity.

## Results and discussion

As summarized in Fig. 1, and comprehensively revised by others[5,6,30–33], experimental efforts to control defect formation in Zr-and Hf-UiO-66 have been mainly concentrated in using the modulation approach. This strategy relies on the addition of an excess of monocarboxylic acids (e.g. formic, acetic or benzoic acid) to the benzene-1,4-dicarboxylic acid (BDC) linker present in the reaction mixture. Monocarboxylic acids can compete with BDC for the coordination points in the cluster to control framework assembly and to enable the formation of defective structures (either ML or MC). Figure 1 shows a chemical space in which defect generation is controlled by three main synthetic variables: linker to metal ratio (L/M), modulator-to-linker ratio (Mod/L) and reaction temperature (T). Previous results reveal a shocking tendency towards the use of L/M ratios close or above 1.0, more likely to favor the formation of non-defective 12-c *fcu* UiO-66 phases ($M_{24}L_{24}$), compared to 0.66, that would be arguably required to form missing cluster *reo* phases ($M_{18}L_{12}$). A notable, and maybe the only exception to this rule, is the well-known report by Cliffe et al.[24], where the use of sub-stoichiometric Hf/L = 0.2 ratios led to the formation of the *reo* phase with structured diffuse scattering, characteristic of spatially correlated defects. Figure 1 also reveals the poor attention dedicated to temperature as synthetic variable in this context. Even though temperature was early reported to be determinant for the appearance of superlattice powder X-ray diffraction (PXRD) lines characteristic of correlated *reo* domains[22], this parameter has been mostly disregarded in the literature. Overall, the library of representative conditions in Fig. 1 suggests a point-by-point sampling of the composition (M, L, and Mod) and temperature space. Compared to inefficient blind exploration, we argued that systematic synthesis and characterization routines would be more adequate to navigate this complex chemical space and identify the leading variables that control the formation and spatial correlation of defect domains in UiO-66. This approach is built upon our successful results on the synthesis of elusive titanium and zirconium frameworks[34–37]. Based on previous results[30], we

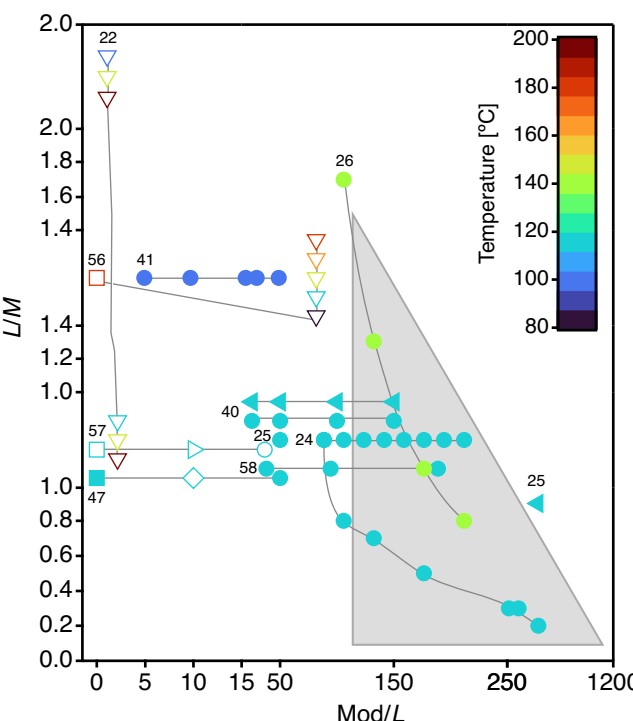

**Fig. 1 | Summary of the experimental conditions used by different authors to produce UiO-66 structures with missing cluster defects.** Data selection is based on the bibliographic survey by Feng et al.[30]. Please note that, for clarity, axis do not follow a linear scale. *y* and *x* axis display respectively, the linker/metal (L/M) and modulator/linker (Mod/L) ratios used in all works selected. Modulators: FA (formic acid), TFA (trifluoroacetic acid), AA (acetic acid) and BA (benzoic acid). Solv refers to the solvent used during the synthesis. Linkers used were either benzene-1,4-dicarboxylic or 2-aminobenzene-1,4-dicarboxylic acids and metal salts ZrCl₄, ZrOCl₂ or HfCl₄. The chemical space explored in this work is delimited with a gray triangle. We refer the reader to Supplementary Table 1 for a detailed description of the experimental conditions, corresponding references are signaled with small numbers[56–58].

decided to use *N,N*-Dimethylformamide (DMF) as solvent and HfCl₄ as metal source, as its higher atomic scattering factor for X-rays compared to Zr would facilitate analysis. As far as the modulator is concerned, formic acid (FA) seemed the best choice to minimize cross competition with the FA that would be formed from the decomposition of DMF in acid medium during the reaction. Moreover, and to minimize concentration effects, all experiments were designed to maintain the amount of reagents ($n_{HfCl_4} + n_{BDC}$) and the volume of solvent ($V_{DMF} + V_{FA}$) constant thorough all the synthesis. We decided to react our mixtures 48 h, which is the maximum reaction time used in previous reports and would be adequate to avoid, as far as possible, the formation of kinetic products. We used a FLEX SHAKE HT workstation from Chemspeed for robotic dispensing of solids and liquids combined with 96-well PXRD and 25-well thermogravimetric analysis (TGA) plates. This helped to ensure high reproducibility and accelerate the systematic screening of reaction variables based on the characterization results.

### Systematic exploration of temperature and (L/M/Mod) chemical space

In a first series of experiments, labeled as Series A (Supplementary Table 2 and Supplementary Fig. 1), we reacted variable amounts of HfCl₄ (M) and BDC (L) in 4:1 v/v DMF/FA mixtures at 100, 120, 140, and 160 °C for 48 h in 25 mL Teflon bottles. We fixed $n_{HfCl_4} + n_{BDC}$ to 0.6 (0.12 M total concentration) and the temperature, and varied

systematically L/M between 0.1 and 1.5 for a total of more than 28 reactions divided in 4 subsets. Importantly, this experimental design involves changes in headspace pressures within the reaction vessels as a function of reaction temperature and solvent mixture. The headspace pressure increase with temperature is difficult to quantify, but the reader should keep in mind throughout the text that when we refer to a temperature modification, we are truly referring to a modification of the reaction temperature and the associated change in the pressure inside the reactor. All reaction products where isolated as white solids in at least some tens of milligrams, except for the 100 °C set that led to significantly smaller recovered masses regardless the L/M ratio used, indicative of a less favorable MOF formation at this temperature. PXRD patterns as a function of the temperature for Series A are summarized

in Supplementary Fig. 2. Only low temperature samples (T ≤ 120 °C) showed appreciable superlattice reflections at low angular values that might be indicative of correlated defect domains in these solids, compared to the featureless diffraction patterns of the high temperature samples in the same region. Whereas, stoichiometric combinations (L/M = 1) only show low angle weak and broad Bragg peaks, that are not visible for ratio 1.5, these reflections turn more intense and defined when reducing the amount of linker below 1.0 until reaching a maximum for 0.1 (Fig. 2a, Supplementary Fig. 3). These results are consistent with previous findings by Cliffe et al., that anticipated how low linker compositions are more likely to generate superlattice reflections[24], and suggest the formation of MC Hf$_6$ *reo* domains that grow in size and correlation length for the formation of at decreasing

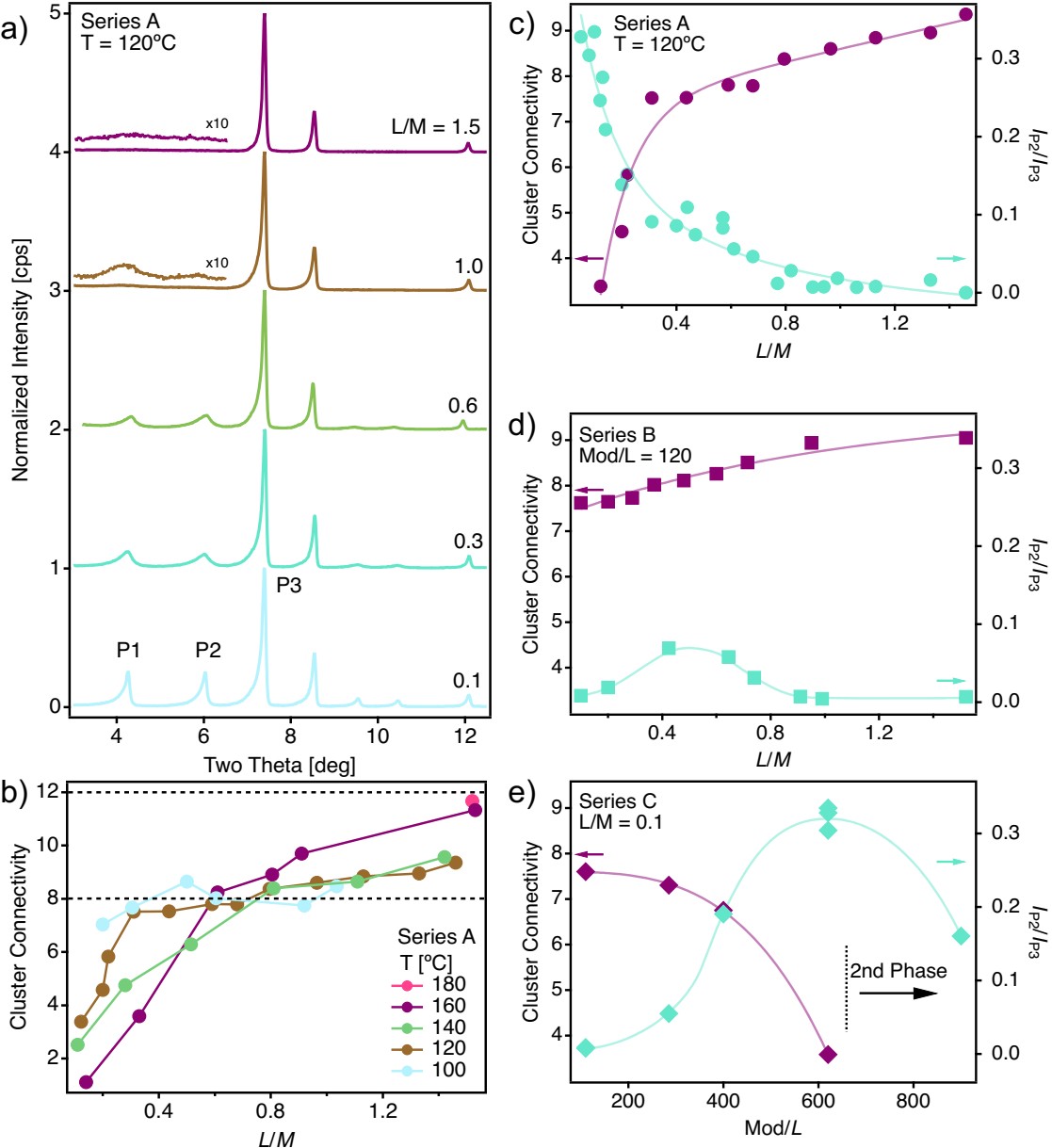

**Fig. 2 | X-ray superlattice diffraction and cluster connectivity of Hf-UiO-66 in series A, B, and C. a** PXRD patterns at low angular values of Series A showing the increase in intensity of superlattice diffraction concomitant to the use of lower linker to metal (L/M) ratios. **b** Compositional analysis calculated from TGA data showing the impact of temperature in the resulting cluster connectivity. Horizontal lines show the theoretical connectivity of Hf$_6$ clusters in ideal *fcu* (12-c) and *reo* (8-c) topologies. Correlation between cluster connectivity and relative

contribution of superlattice diffraction to the PXRD pattern at a temperature (T) of 120°C and 0.1 ≤ L/M ≤ 1.5 calculated from the relative intensities of reflections P2 and P3 ($I_{P2}/I_{P3}$) for: **c** variable linker (L) and modulator (Mod) concentrations (series A, circles), **d** fixed Mod concentration (series B, squares) and **e** fixed Mod concentration (series C, diamonds). Note that the formation of secondary crystallographic phases at Mod/L > 600 prevented a meaningful compositional analysis for these samples.

L/M ratios. To confirm this point, we used TGA to correlate synthetic variations with changes to the average cluster connectivity, which from this point on we will refer to as cluster connectivity for brevity's sake. TGA plots show decomposition profiles that can be separated in three stages between 20–200, 200–300, and 500–700 °C (Supplementary Fig. 4). While the first and last have been consistently associated to the loss of physisorbed solvent molecules and framework decomposition, the second has been attributed by different authors either to the loss of modulator[38], DMF guest molecules[17], or the thermal decomposition of free linkers occluded in the pores of the framework[22]. Our thermobalance with mass spectrometer (TGA-MS) data confirm the first and third stage assignations and support that in our case, the second step can be attributed to the combined loss of coordinated FA and DMF guest (Supplementary Figs. 5 and 6). Moreover, it reveals how the relative contributions of FA and BDC to the overall mass loss and MS profiles are inversely proportional, supporting that missing linker defectivity is concomitant to the incorporation of FA to the framework.

According to TGA-MS data (Supplementary Figs. 5 and 6), the third weight loss is well suited to estimate the number of linker molecules per cluster or cluster connectivity number (CN, Supporting Table 5). As shown in Fig. 2b and c, Hf-UiO-66 with CNs near to 12 (defect-free *fcu* topology) can only be obtained for L/M values above the stoichiometric ratio and temperatures above 160 °C. This effect of temperature in preventing the creation of missing linker vacancies has been reported[22], and was here confirmed with an additional sample prepared at 180 °C and L/M = 1.5 for an slight decrease in linker defect concentration and CNs close to ideal 12-c *fcu* frameworks. Reactions at lower temperatures or linker compositions always triggered a drastic increase in linker defect concentration for a reduction in the cluster connectivity, that leads to CNs near ideal 8-c *reo* frameworks, and below for very low L/M ratios. Overall, this compositional analysis suggests a complex scenario in which the generation of ML and MC vacancies in Hf-UiO-66 cannot be dissociated regardless the reaction temperature. However, X-ray diffraction shows that this defect concentration is translated into correlated domains that result from $Hf_6$ MC vacancies, exclusively for T ≤ 120 °C and low linker concentration samples, as supported by the appearance of superlattice reflections with minimum impact to the crystallinity of these frameworks (Supplementary Fig. 2). To correlate MC order with linker concentration, we quantified the relative contribution of this reflections to the PXRD pattern as the ratio between the intensities of the second (P2) and third (P3) peaks for the 120 °C set (Supplementary Note 1). Figure 2c shows how the intensity of superlattice reflections associated to the presence and growth of ordered MC defect domains is concomitant to the reduction in the cluster connectivity, and is drastically maximized for low L/M ratios. Our first set of experiments seem to confirm a clear control of low temperatures and L/M ratios over the formation of correlated defect domains. However, the experimental design of Series A requires a fixed amount for $n_{HfCl_4} + n_{BDC}$ and $V_{FA} + V_{DMF}$, thus imposing variations to the corresponding Mod/L ratios between 95 and 620 for the whole L/M compositional space explored (Supplementary Table 2 and Supplementary Fig. 1). As this constraint did not enable us to extract the individual role of modulator concentration over defect concentration, we designed a second series of experiments (Series B, Supplementary Table 3 and Supplementary Fig. 8) at 120 °C for sampling the same L/M variations at a fixed modulator concentration (Mod/L = 120) by adjusting the relative volumes of FA and DMF. As shown in Fig. 2d, the same analysis used for series A reveals now a minimum impact of L/M over the cluster CN that remains near 8 ($Hf_{18}L_{12}$) in all cases. In turn, superlattice diffraction is only present for L/M ratios close to the 0.66 ideal *reo* ratio. Thus, the formation of correlated defect domains seems to be favored for high Mod/L, possibly due to the favored metal complexation with FA in competition with BDC molecules for the coordination of the ($Hf_6$) clusters during

frameworks assembly. For sampling the effect of systematic variations of the modulator concentration in the interval 110 < Mod/L < 1100 at constant L/M ratios between 0.1 and 0.8 and the same temperature, we designed Series C (Supplementary Table 4. Supplementary Fig. 9 shows that increasing Mod concentration at a fixed L concentration lead to a progressive increase in the intensity of the superlattice reflections in the diffraction pattern up to a critical value, which changes with the linker concentration, and that triggers the formation of a secondary crystallographic phase which eventually becomes the only one present. We tentatively assign this secondary phase to HfFA (Supplementary Fig. 10), that is an isoreticular analogue of the framework ZrFA built from $Zr_6O_4(OH)_4$ clusters and formate linkers[39]. This mixed-phase complex scenario prevented TGA compositional analysis for the whole Mod/L interval and was restricted to the single-phase samples. Data for the L/M = 0.1 subset (Fig. 2e) shows a clear effect of the modulator concentration over cluster CN and the corresponding increase in the intensity of superlattice reflections, characteristic of the formation of correlated *reo* domains, for a maximum contribution to the diffraction pattern at Mod/L = 600.

## Defining the defect engineering zone in UiO-66 frameworks

Once we have established that correlated defect domains where only present in low temperature samples (T ≤ 120 °C), the global analysis of data extracted from series A, B and C at 120 °C was finally used to delimit the hot area within the compositional space defined by L/M and Mod/L that is better suited to gain chemical control over the formation of correlated defect domains in Hf-UiO-66 in terms of superlattice diffraction (Fig. 3a) and cluster connectivity (Fig. 3b). These compositional plots account for the chemical space delimited in Fig. 1 and were calculated using WaveMetrics Igor Pro 9 through Voronoi interpolation of the Delaunay triangled diffraction and TGA data for all samples available (Suplementary Note 2). Samples falling outside the colored map correspond to either the formation of secondary crystallographic phases (open circle symbol) or to the inability to form UiO-66 like solids (cross symbol), thus suggesting the critical effect of relative L/M and Mod/L ratios, not only in controlling defect formation and correlation, but even to enable the assembly of the framework. We observe a positive effect over the formation of correlated defect domains and corresponding superlattice diffraction peaks at high Mod/L ratios because of the maximized competition of FA with BDC for metal complexation and reduced cluster CN. However, increasing FA concentration imposes a reduction in the DMF fraction in the reaction media, which causes that at low solvent fractions UiO-66 phase is no longer dominant. Only for low L/M ratios, high Mod concentrations can be made compatible with high solvent fractions for the formation of almost crystallographically pure defective UiO-66 phases. Our data also reveal the prevalence of clusters with connectivity 8 regardless the experimental conditions used. In fact, 12-c $Hf_6L_{12}$ clusters indicative of ideal *fcu* topologies can be produced only at very limited conditions, in which the use of high temperatures (T ≥ 160 °C) and an excess of BDC (L/M = 1.5) above the stoichiometric ratio are both required to overcome the thermodynamic preference of this cluster for the formation of vacancies. As shown in Fig. 3c, compared to the well-defined conditions required for the formation of defect-free UiO-66 samples, we hypothesize that at lower temperatures the formation of random ML vacancies takes place continuously from high L concentrations, and is responsible for the generation and growth of *reo* domains with spatial correlation. Within this area, the combined effect of Mod and L concentrations can be used to chemically control their relative proportion within the framework that is maximized for high Mod/L values. For low Mod/L (Mod/L ≤ 400) MC correlation can be maximized when L/M approaches 0.5, a value close to *reo* phase stoichiometry (L/M = 0.66). In this case neither L or M in solution are in excess with respect to each other and network expansion during crystal growth is slow enough to allow some degree of defect

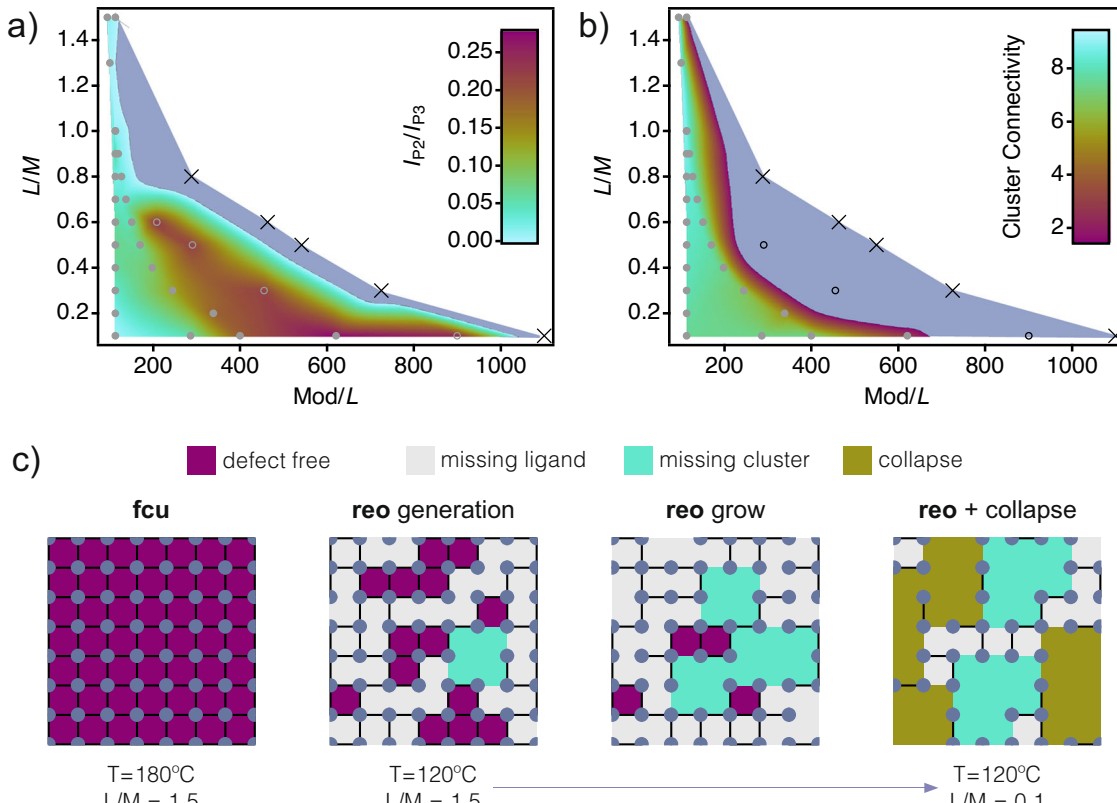

**Fig. 3 | Defined area for chemical control of defects and proposed mechanism for generation and growth of correlated domains in Hf-UiO-66.** Compositional plots interpolated from Series A, B, and C data at a temperature (T) of 120 °C. The color-z-axis used respects the numerical value of the experimental point used in each case. **a** Correlation of superlattice diffraction and **b** Cluster connectivity as a function of the linker to metal (L/M) and modulator to linker (Mod/L) ratios.

Samples in the gray area have not been considered for analysis. Open circle and cross symbols correspond to the formation of secondary phases and the inability to form a UiO-66-like solids, respectively. **c** Proposed structural description for the different stages in the formation and growth of **reo** domains starting from defect-free **fcu** topology and accounting for the variations in superlattice diffraction and cluster connectivity observed for variable L/M ratio.

correlation. On the other hand, spatial correlation can be imposed at high modulator and low linker concentrations at the expense of a dramatic loss of cluster connectivity well below the ideal CN = 8. This is arguably the reason for which the appearance of superlattice diffraction in defective samples is always accompanied by a drastic reduction of overall cluster connectivity, with CNs near 4 for the most defective ones. This is far from ideal CN = 8 for **reo** topology and suggest an interplay between ML and MC vacancies that cannot be disentangled during the formation of correlated defect domains. Despite the reported tolerance of this cluster to accommodate ML vacancies[8], we argue that this progressive reduction in connectivity in the final stages of **reo** growth will ultimately lead to framework instabilities, that might provoke structural collapse in the domains dominated by CN < 4.

### Effect of reo domain formation on framework structure and porosity

Although our systematic analysis reveals useful trends for controlling the formation of defect domains, the coexistence in the same solid of two crystallographic phases sharing some diffraction peaks makes it difficult to provide an accurate quantification of their relative fractions or capture the effect that the formation of these domains might have over the porous structure of the defective frameworks formed at final and intermediate stages proposed in Fig. 3c. To confirm this point, we analyzed in more detail a sub-set of defective samples (Series A, T = 120 °C and 0.1 ≤ L/M ≤ 1.5) and our reference for an ideal defect-free **fcu** framework (T = 180 °C and L/M = 1.5). As already anticipated from our simpler peak ratio analysis, Rietveld refinement of the high-quality diffraction data of these samples reveal a progressive increase in the

relative fraction of the **reo** phase ($\chi_{reo}$) to the overall diffraction pattern with decreasing BDC concentrations (Fig. 4a, Supplementary Tables 6 and 7, Supplementary Figs. 12–14). The structural analysis reveals that control of temperature and L/M ratios can indeed be used effectively for producing **fcu** and **reo** phases at either above or below the stoichiometric amounts required for forming UiO-66. Figure 4b, c shows the Rietveld refinement and the corresponding structures for both phases generated from experimental data. To the best of our knowledge this is the first experimental crystallographic report for the **reo** phase, which has been typically represented from computational simulations. The corresponding structural file have been made available through the Cambridge Crystallographic Data Centre (CCDC number 2223895). The refinement of linker occupancies in the structure is also consistent with the prevalence of the clusters with connectivity 8 at T = 120 °C described above. Compared to samples at 120 °C, T = 180 °C sample display a cluster CN close to 11, in better agreement with the ideal framework connectivity of **fcu** phase (Fig. 4a). We observe a drastic deviation of the CN calculated from TGA and Rietveld analysis for the most defective samples with $\chi_{reo}$ near 50% or above (L/M ≤ 0.6). This agrees well with the formation of structurally collapsed domains at low linker concentrations proposed above, that would do not contribute to Bragg diffraction in the X-ray diffraction pattern and can be only captured by chemical analysis of the bulk.

To further confirm this point, we collected $N_2$ isotherms at 77 K for all these solids in a 3Flex analyzer equipped with high-resolution micropore ports. The corresponding adsorption-desorption profiles and pore size distribution (PSD) plots calculated with the NLDFT kernel fitting method available, are shown in Fig. 5a and b, respectively. Our

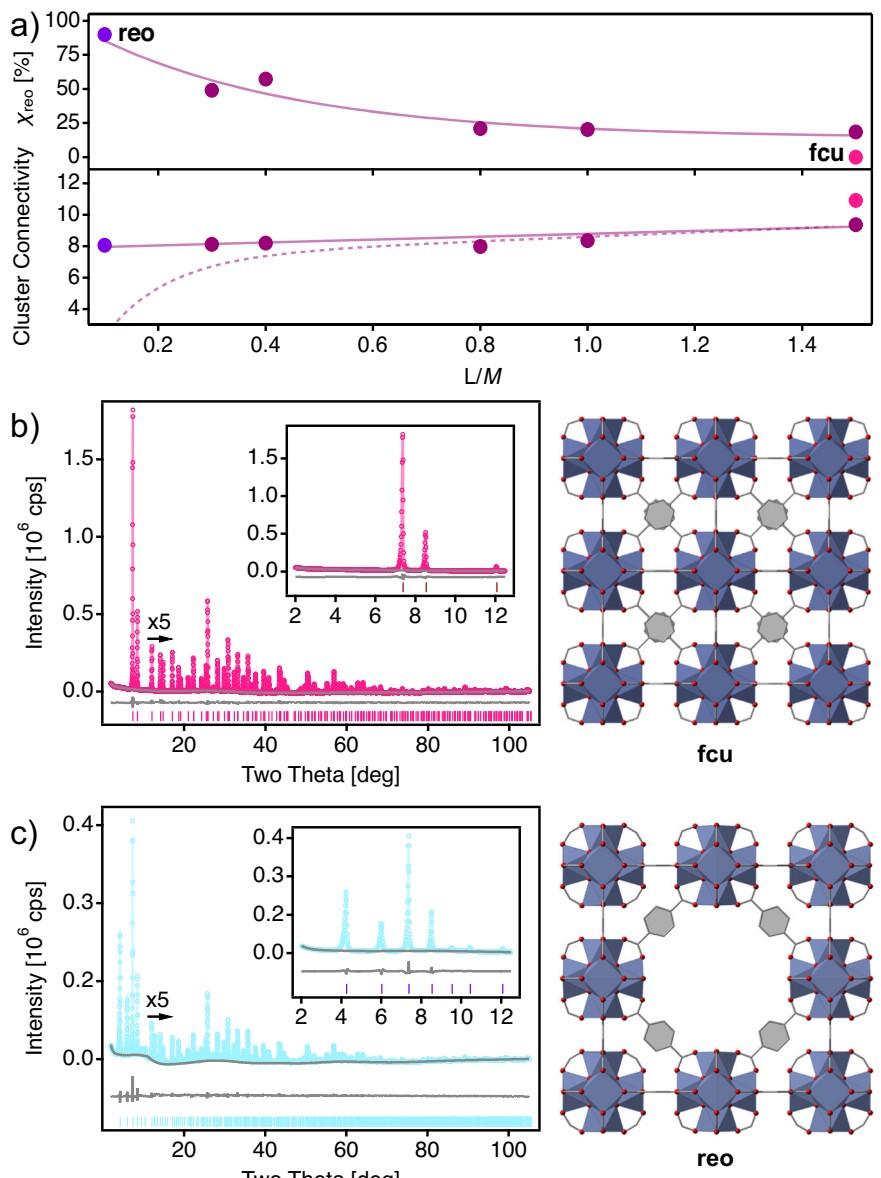

**Fig. 4 | Rietveld refinement and relative fraction of *reo* domains in defective Hf-UiO-66. a** Top: correlation of the relative fraction of the *reo* phase ($\chi_{reo}$) percentage calculated from the Rietveld refinement of samples produced at varying linker to metal (L/M) ratios. *Bottom*: comparison of the evolution in cluster connectivity determined by refinement of linker occupancies (solid line) and TGA compositional analysis (dashed line). Grape and maroon dots correspond to near ideal *reo* and *fcu* crystallographic phases. Solid lines are a guide to the eye. Rietveld refinement and experimental structures of the **b** 12-c *fcu* and **c** 8-c *reo* phases of Hf-UiO-66. Experimental (hollow dots), calculated (solid line), difference plot [($I_{obs}$-$I_{calc}$)] (line, bottom panel) and Bragg positions (ticks, bottom panel). Atoms are represented in color: zirconium (blue polyhedron), oxygen (red), carbon (gray sticks). See Supplementary Table 7 and Figs. 13 and 14 for more details on the refinement of all samples.

reference for a nearly ideal *fcu* framework (CN = 11) shows an uptake slightly above the theoretical value estimated with Zeo++ for a defect-free structure[38,40]. This deviation agrees well with the effect of missing linker vacancies in $N_2$ uptake reported in previous works[25,41], and is consistent the corresponding PSD curve, that confirms the presence of the tetrahedral (0.85 nm) and octahedral (1.1 nm) microporous cavities characteristic of this topology (Fig. 5c). The sub-set of samples prepared at 120 °C that display a low $\chi_{reo}$ fraction (L/M = 1.5, 1.0 and 0.8) show much higher pore volumes up to 406.7 $cm^3 \cdot g^{-1}$, well above the values expected for an ideal *reo* framework. According to the PSD plot, this boost can be arguably attributed to the combination of residual *fcu*-type porosity and the generation of bigger pores (near 1.6 nm) in these samples, characteristic of the MC vacancies intrinsic to *reo* domains (Fig. 5d). This is followed by a drastic gas uptake reduction to

values in between both topologies for the samples with $\chi_{reo}$ > 50% (*ca.* 0.4 and 0.1), which is accompanied by a complete loss of *fcu* contribution to the PSD that is mainly dominated by *reo*-type porosity. As anticipated by the very low cluster CNs for these set of samples, determined by TGA compositional analysis (Fig. 4a), we argue that the structural collapse of a fraction of the framework might be the main responsible for this acute reduction in accessible pore volume. It is worth nothing that without the use of specialized techniques as positron annihilation lifetime spectroscopy (PALS), small-angle scattering (SAS) or Krypton physisorption, it is difficult to determine if this pore volume reduction is associated to the presence of inaccessible pore regions inside the structure due to local collapse or it is caused by the total collapse of some domains in the solid. It is worth noting that structural collapse is correlated with the onset of a well-defined

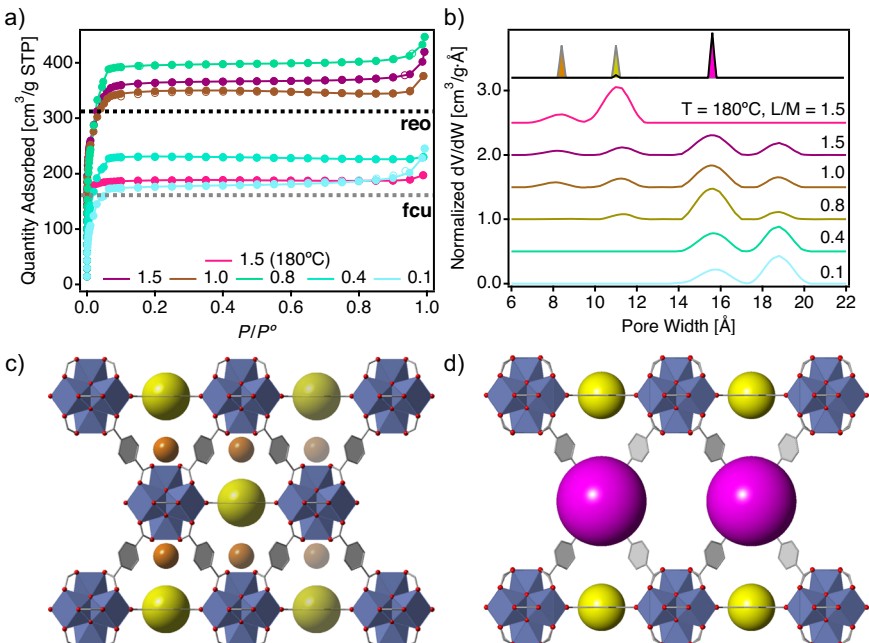

**Fig. 5 | Correlation of defect generation, growth, and collapse with gas uptake.** **a** $N_2$ isotherms at 77 K of the defect-free sample used as a reference (linket to metal ratio (L/M) = 1.5, temperature (T) = 180 °C) and defective samples synthesized at 120 °C for varying linker concentrations, the horizontal axis is the relative pressure (P/P°). Adsorption and desorption branches are represented by filled and empty symbols. Dotted lines correspond to the expected values for ideal ***fcu*** and ***reo*** topologies calculated with Zeo[++ 38,40,52]. **b** Pore size distribution plots calculated with a NLDFT kernel for the experimental isotherms and normalized to the total

accessible volume so that ∫(dV/dW) dW = 1 for all distributions. Pore width (W) values for the tetrahedral (orange) and octahedral (yellow) cavities intrinsic to the ***fcu*** framework and the bigger cavities (purple) formed in ***reo*** topologies from missing cluster vacancies have been added as a reference. Crystallographic views of **c**) ***fcu*** and **d**) ***reo*** frameworks along the [100] direction for identification of these cavities. Atoms are represented in color: zirconium (blue polyhedron), oxygen (red), carbon (gray sticks).

---

contribution to the PSD near 1.9 nm, that gains relative weight in the PSD but cannot be ascribed to any of the crystallographically pure phases. Our porosity data are overall consistent with the mechanism proposed in Fig. 3c. Defectivity is incorporated by forcing ML vacancies, which can maximize the probability of generating neighboring MC vacancies for spatially correlated ***reo*** domains and consequent increase in pore volume, but also favor the formation of $Hf_6$ clusters with very low connectivity index for mechanically unstable domains more prone to collapse for a drop in porosity. To confirm that this effect was triggered by partial sample collapse and not the formation of hafnium oxide phases for low linker concentrations, we performed morphological scanning electron microscopy (SEM) characterization and full stoichiometric determination of all samples with energy dispersive X-ray spectroscopy (EDX) of the solids and nuclear magnetic resonance (NMR) after basic digestion (Supplementary Figs. 16 and 17)[42]. The octahedral shape of the crystals is maintained in all cases (Supplementary Fig. 15). The local cluster connectivity always remains near to 12 (Supplementary Fig. 17), and we observe how the decrease in linker concentrations triggers a removal of BDC linkers, which is counterbalanced by the introduction of FA capping ligands to the $Hf_6$ nodes to complete all the coordination positions available.

## Imaging of defect domains with high-resolution electron microscopy

Spherical aberration scanning transmission electron microscopy (Cs-corrected STEM) was used to analyze the different Hf-UiO-66 frameworks obtained. Because of the near octahedral morphology of Hf-UiO-66 crystals, all the STEM data was collected along [110] zone axis as it was found to be the preferential orientation for this morphology. The Cs-corrected STEM annular dark field (ADF) images and the Fast Fourier Transform (FFT) obtained from the ADF images of the defect-free sample used as a reference (L/M = 1.5, T = 180 °C) and defective

samples synthesized at 120°C for varying linker concentrations are shown in Fig. 6a–h and Supplementary Figs. 18–22. To visualize the fine structure of the non-defective and L/M = 0.1 samples we turned into Annular Bright Field (ABF) imaging (Fig. 6i, j). ABF has been proved to be a useful approach for the analysis of light elements in beam sensitive materials including MOFs[43,44]. For the non-defective sample, the crystals displayed a well-defined octahedral morphology with very sharp edges where the $Hf_6$ nodes correspond to the signal with the strongest contrast (Fig. 6a). The FFT obtained from the ADF image can be indexed within $Fm\bar{3}m$ space group that matches with the simulated electron diffraction (ED) for the ***fcu*** structure (Fig. 6e). The observation of the fine structure was achieved with ABF imaging, see Fig. 6i (for clarity, the contrast was inverted). In here, it is possible to differentiate even the two atomic $Hf_6$ columns (brightest spots) as well as the organic linkers (lower contrast observed between the metal clusters). To facilitate image interpretation, the structural model and the simulated image (white rectangle) were overlaid over the ABF image, observing a perfect concordance with the experimental data. Interestingly, at first sight no significant structural differences were observed for the sample synthesized at 120 °C (Fig. 6b), the morphology and crystal structure were very similar to the non-defective one. However, the FFT presented less diffraction spots (Fig. 6f), indicative of the lower spatial resolution achieved associated to the lower stability under the e-beam. This lower beam stability, compared to the non-defective sample, may be associated to the formation of ML vacancies that were not observable due to the low stability of the material. The FFT could be also indexed assuming a $Fm\bar{3}m$ space group, indicating the predominance of ***fcu*** domains. The extra weak signals (white circles in Fig. 6f) can be explained by the formation of a small number of sparse MC defect domains, in accordance with a sample with cluster connectivity close to 8 (Supplementary Fig. 23). Significant differences were observed for samples synthesized at

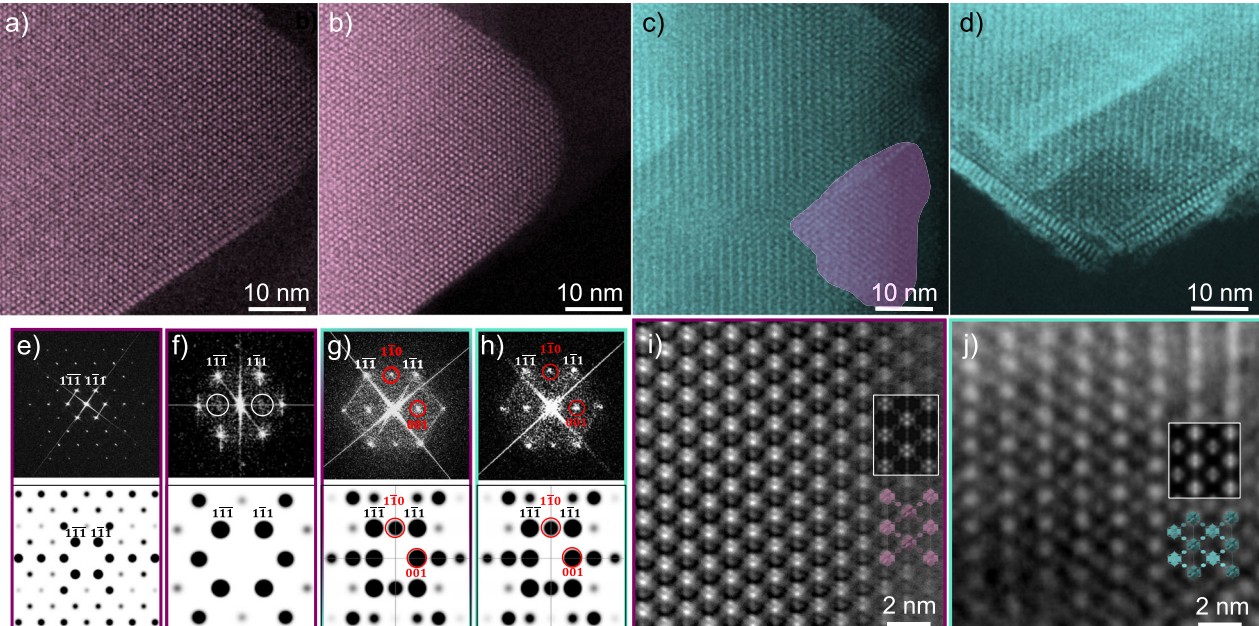

**Fig. 6 | Cs-corrected STEM analysis of Hf-UiO-66.** High-resolution STEM-ADF images of **a** the defect-free sample used as a reference (linket to metal ratio (L/M) = 1.5, temperature (T) = 180 °C) and defective samples synthesized at 120 °C and L/M = (**b**) 1.5, (**c**) 0.4 and (**d**) 0.1. *fcu* and *reo* domains correspond to magenta and turquoise zones in these images, respectively. **e–h** From left to right large area FFT and simulated ED patterns corresponding to (**a–d**), respectively. Spots forbidden in the $Fm\bar{3}m$ space group have been signaled with red circles; same images without scaling can be found in Supplementary Fig. 22. High-resolution STEM-ABF images with the contrast inverted of the **i**) reference (**a**) and **j**) L/M = 0.1 sample (**d**), structural models and the simulated images (white rectangle) have been overlaid over the ABF image. The almost negligible *reo* spatial frequencies in (**f**) have been signaled as (white circles), small area FFTs of (**b**) and (**c**) can be found in Supplementary Figs. 23, 24, respectively. All images have been denoised using HREM filters implemented in Digital Micrograph[54].

120 °C with L/M = 0.4 and L/M = 0.1. ADF image of L/M = 0.4 (Fig. 6c) show areas with different contrast in the distribution of the Hf clusters (magenta and turquoise areas), suggesting that a structural variation took place in the turquoise zone. Furthermore, the FFT (Fig. 6g), presented additional spots that were forbidden in the $Fm\bar{3}m$ space group, that however, could be indexed within the $Pm\bar{3}m$ space group as 001 and $1\bar{1}0$ reflections that are in agreement with the simulated ED from the *reo* phase, highlighting the co-existence of *fcu* and *reo* domains in the same crystal (Supplementary Figs. 18–24). Figure 6d corresponds to a region of a L/M = 0.1 sample whose structure corresponded to pure *reo* phase. The FFT (Fig. 6h) can be unambiguously indexed as $Pm\bar{3}m$ space group (also in agreement with the simulated ED patterns). The high-magnificated ABF micrograph is presented in Fig. 6j. In this case, the strong contrast corresponded to the $Hf_6$ clusters while the weaker contrast is directly derived from the MC. Again, the structural model and the simulated image for the *reo* phase have been overlaid, in good agreement with the experimental image. It is important to stress out that the materials with missing units, linkers and specially clusters, present lower amount of diffraction spots in the FFT, and much lower e-beam stability, which did not allow the visualization of the organic linkers. (Fig. 6e–h).

### Effect of defect concentration on catalytic performance

Defective UiO-66 frameworks have been thoroughly explored for catalytic applications[15,31,45]. The presence of defects has been shown to improve catalytic performance by increasing the concentration of catalytic active sites or overcoming substrate diffusion limitations[6,30]. As most of these studies are based on Zirconium(IV) frameworks, we decided to synthesize a family of defective Zr-UiO-66 analogues by substituting $HfCl_4$ for $ZrCl_4$ as the source metal salt, and following exactly the same synthetic conditions previously described for Series A samples. The properties of both families are closely related, confirming that the synthetic principles previously discussed can be straightforwardly transferred from Hf to Zr (Supplementary Figs. 25–29).

Regarding superlattice diffraction and cluster connectivity, both series are almost indistinguishable with Zr samples showing a slightly higher tendency to display more intense superlattice reflections at low L/M values, which might be indicative of the formation of MC domains with larger correlation lengths (Fig. 7a). We chose the epoxide ring-opening amination reaction between aniline and cyclohexene oxide as a model catalytic test (Fig. 7b) to evaluate correlations between defect density and catalytic activity in our samples[41,46–48]. As shown in Fig. 7c and additional supporting Figure, catalytic performance is highly dependent on the defect concentration of the samples. Compared to the negligible activity of the defect-free *fcu* reference sample (brown circles) and control sample (black circles), the introduction of defects in the samples synthesized at 120 °C increases the activity and reach full conversion for L/M = 0.6 after 10 h at 30 °C (10% mol catalyst). This catalytic activity is near the benchmark performance reported for the 8-c nodes in MOF-808 capped with triflate ligands[49]. From this point, higher defect concentration in L/M = 0.1 provokes a drop-in activity, likely associated to the structural collapse of domains with very low cluster CN and for a consequent drop in porosity an active site inaccessibility, as observed for the Hf analogue (Fig. 5a).

### Concluding remarks

Summing up, by using UiO-66 as an experimental platform, we demonstrate that periodic missing cluster vacancies can only take place at temperatures ≤ 120 °C, and can be chemically controlled by the relative concentrations of metal, linker, and modulator. Whereas the formation of random missing linker vacancies takes place in a broad compositional space, the onset of structurally correlated defect domains is only accessible for L/M ratios near or below the ideal stoichiometry of the 8-connected *reo* phase (L/M = 0.66). Gradual decrease of this experimental ratio permits synthesizing defective samples with increasing correlated disorder, that is here used to obtain almost crystallographically pure *reo* phases. The comparative analysis of these samples by using Cs-corrected STEM imaging provide direct

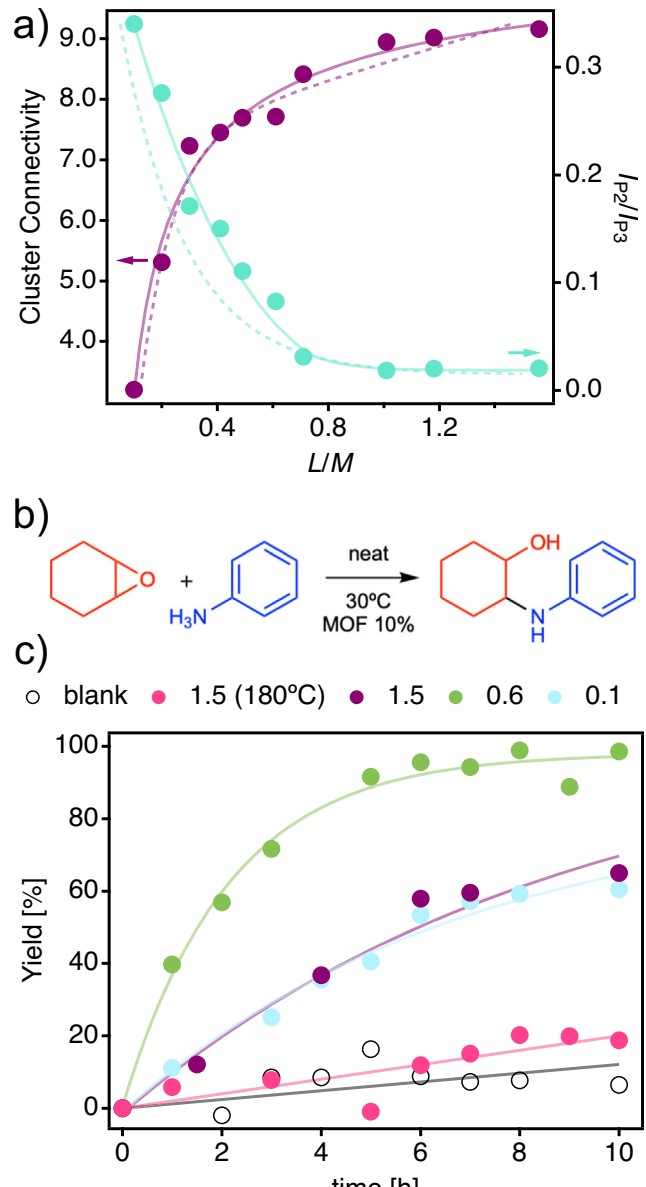

**Fig. 7 | Defect control of catalytic activity. a** Correlation between cluster connectivity (magenta) and relative contribution of superlattice diffraction ($I_{P2}/I_{P3}$, turquoise) to the PXRD pattern for Zr-UiO-66 samples synthesized at a temperature (T) of 120 °C. Experimental trends for Hf analogues are represented by dashed lines. **b** Scheme of the epoxide ring-opening reaction used to test the catalytic activity. **c** Kinetic profile of the defect-free sample used as a reference (L/M = 1.5, T = 180 °C) and defective samples synthesized at 120 °C for varying linker to metal (L/M) ratios. Open circles correspond to a control sample in absence of MOF. Lines are a guide for the eye.

proof of the progressive formation of structurally ordered nanodomains of missing linker and cluster vacancies, whose relative sizes are fixed by the linker concentration. Additional analysis of the porosity and catalytic activity of these defective samples reveals that, whereas missing cluster vacancies have a positive effect at relative crystal fractions near $\chi_{reo} = 50\%$, surpassing this value have a detrimental effect in pore volume and activity as result of the partial collapse of the framework. We are confident that our systematic experimental design and characterization routines holds great promise to gain synthetic control over correlated disorder in reticular frameworks. The possibility of translating synthetic coordinates into specific concentrations and distributions of defects at a single crystal level is arguably the next

step for transforming defects into an advanced tool for the design of more complex frameworks with advanced functions.

## Methods

### Materials and reagents
$ZrCl_4$ (Aldrich, 99%), $HfCl_4$ (Aldrich 98%), benzene-1,4-dicarboxylic acid (BDC, Aldrich, 98%), formic acid (FA, Alfa Aesar, 97%), N,N-dimethylformamide (DMF, Scharlau), aniline (>98 TCI Chemicals), cyclohexene oxide (Thermo Scientific, 98%), n-dodecane (abcr Chemicals, 99%) and all the solvents were used as received without further purification.

### Synthesis of UiO-66 samples
We used a Flex Isynth HT workstation from Chemspeed for robotic dispensing of $MCl_4$ (M = Hf, Zr) and BDC into VWR 89026-060 PTFE (height, body diameter, neck diameter = 61, 33, 19 mm) bottles ($n_{MCl_4} + n_{BDC} = 0.6$). Next, 5 mL of the desired FA/DMF mixture were added to the bottle. The bottle was sealed and gently shacked before putting it into an oven. Samples were heated to the desired temperature (100 −160 °C) at 10 deg/min and reacted for 48 h. After this time, they were naturally allowed to cool down to room temperature. The solvent was removed by centrifugation (1 min, 515 rcf) using a Eppendorf 5804 centrifuge and the solids washed with DMF (3 × 5 mL, 1 min, 515 rcf) and acetone (1 × 5 mL, 1 min, 515 rcf). At this point, the solids were either allowed to dry overnight or where solvent-exchanged several times with acetone during a period of at least one week. When required, samples were vacuum activated for 16 h at 60 °C and $10^{-6}$ Torr prior to analysis. The synthesis of 180 °C samples, not supported by our Teflon reactors, was carried without the aid of the robot and using a 25 mL PTFE-lined Parr reaction vessel. For Series A samples we used FA/DMF = 4/1 v/v mixtures, for Series B FA/BDC molar ratio was 120, for series C both L/M and FA/BDC was changed along the series (see Supporting Tables 2–4).

### Physical characterization
X-ray diffraction (XRD) patterns were collected in a PANalytical X'Pert PRO diffractometer using copper radiation (Cu K$\alpha$ = 1.5418 Å) with an X'Celerator detector, operating at 40 mA and 45 kV in the 3° < 2θ < 40° range with a step size of 0.017°. Samples were mounted on a 96-well PXRD plate. Powder X-ray diffraction (PXRD) patterns for refinement were collected on solvent exchanged activated samples using a 0.5 mm glass capillary mounted and aligned in a PANalytical Empyrean diffractometer using copper radiation (Cu K$\alpha$ = 1.5418 Å) with a PIXcel detector, operating at 40 mA and 45 kV. Profiles were collected using a Soller Slit of 0.02° and a divergence slit of 1/4 in. at room temperature in the angular range 2° < 2θ < 40° with a step size of 0.017°.

Rietveld refinements were carried out over previously activated samples using the software TOPAS Academic v6 [http://www.topas-academic.net/][50]. For the samples with variable L/M ratios, the Rietveld refinements were carried out with considering the presence of both **fcu** and **reo** phases. Prior to Rietveld refinement, a full profile powder refinement was carried with the LeBail method in all the samples to confirm the space group and phase purity. A pure crystallographic phase is defined as a phase in which the position and intensities of all the peaks in a powder diffraction pattern can be assigned to a single crystalline phase. Non-defective **fcu** and **reo** Hf-UiO-66 models with $Fm\bar{3}m$ and $Pm\bar{3}m$ symmetry were created based on previously reported structures [https://www.nature.com/articles/s41557-019-0263-4#Sec9][25], with the aid of Materials Studio (MS) 2017 R2. As first approximation, the position and orientation of $Hf_6$ cluster and of the linker of the **reo**-UiO-66 model were fixed according to the computational model. Bond distance restraints were applied in the organic linker, the initial bond distance values were set as 1.38 Å (aromatic C-C), 1.46 Å (exocyclic C-C) and 1.28 Å (carboxylic C-O). H atoms were placed at the ideal positions with a bond distance constrain of 0.93 Å (C-H) in

the final stages of the refinement. All the constrained bond distances were allowed to refine with the exception of the C-H bond, which was fixed throughout the refinement. For the non-defective *fcu*-UiO-66, the occupancy of the BDC linker was allowed to refine, except for the O atoms of the carboxylic group, assuming the replacement of the linker by water or formate molecules. In the case of the *reo*-UiO-66, only the isotropic atomic displacement parameters were allowed to refine when the calculated weight fraction was below 50 wt.%. For weight fractions higher than 50 wt.% of the *reo*-UiO-66, the atomic coordinates of the cluster and linker atoms were allowed to refine. Residual electron density inside the pores was modeled as water molecules, whose position and occupancies were allowed to refine. The background was fitted with a 24-coefficient Chebyshev polynomial and peak-shapes were modeled with a Thompson-Cox-Hasting pseudo-Voigt profile function. The instrumental parameters were obtained from the LeBail refinement of the pristine non-defective UiO-66 and were used as a reference for the refinement of the remaining UiO-66 samples.

Particle morphologies and dimensions were studied by scanning electron microscopy (SEM) using a Hitachi S4800 scanning electron microscope at an accelerating voltage of 20 kV and 10 µA over samples metalized for 90 s with a mixture of gold and palladium. Single point energy-dispersive X-ray (EDX) data was collected on the same instrument at 20 kV over non-metallized samples.

Gas adsorption measurements were performed ex-situ on solvent exchanged activated samples. Surface area, pore size and volume values were calculated from $N_2$ adsorption-desorption isotherms recorded at 77 K on a Micromeritics 3Flex apparatus. Brunauer-Emmett-Teller (BET) Surface area analysis was performed as described for microporous and mesoporous materials[51]. Pore size distributions (PSD) were estimated by NLDFT methods assuming $N_2$−Cylindrical Pores−Oxide Surface model and using 0.0316 non-negative regularization, which provided a good fit to the experimental data. $N_2$ uptakes were estimated from pore volumes calculated with Zeo++ (volpo, samples 10,000, radius 1.86)[52] from ideal *fcu* and *reo* models used as starting point for Rietveld refinement and assuming that the condensate density is that of the liquid $N_2$ (i.e. 0.808 g/cm$^{-3}$)[53].

Thermogravimetric analysis (TGA) was carried out with a Waters/TA Instruments TGA550 apparatus between 25 and 800 °C under synthetic air (5 °C/min and an air flow of 30 mL/min). Samples were mounted on a 25-positions autosampler. Thermobalance with mass spectrometer (TGA-MS) data was collected by Dr. Ion Such Basáñez at the Servicios Técnicos de Investigación of the Universidad de Alicante using a NETZSCH TGA/STA 449 F5 Jupiter instrument coupled to a quadrupolar NETZSCH Aeolos QMS 403 Quadro mass spectrometer, samples were heated under argon between 25 and 800 °C at 10 °C/min. For nuclear magnetic resonance (NMR) measurements 10–20 mg of the sample were dissolved in a 1 M NaHCO$_3$/D$_2$O solution and spectra were recorded on a Bruker Avance 300 spectrometer[42].

### Electron microscopy

Electron microscopy was carried out in a XFEG Titan low-base equipped with a monochromator, a CEOS probe corrector, which was aligned prior the experiments assuring a spatial resolution on a gold standard sample below 0.8 Å. The microscope was also fitted with the Gatan Tridiem Energy Filter (GIF) and an Oxford EDS detector. The samples were placed onto holey carbon copper grids for their observation. data acquisition and analysis was performed using the Realtime module and the HREM Filters Pro from HREM Research Inc[54]. To reduce e-beam damage the electron dose used was maintained under 700 e$^-$/Å$^2$. Data collection was done using simultaneously an annular dark field detector (ADF, inner-outer angles 25–160 mrad), and an annular bright field detector (ABF, inner-outer angles 5–25 mrad). Image simulations were performed using QSTEM software[55] based on

the multislice simulation method (slice thickness 7.4 Å). For that, two different supercells for the fcu and the reo phases were built with dimensions of $124.161 \times 125.97 \times 81.08$ Å$^3$ introducing same microscope parameters as experimentally used: 300 kV, convergence semiangle 15 mrad and aberrations as those obtained after correction (Cs ≈ 700 nm). The thickness of the crystals was based on its similarity to the experimental observation being 5–7 nm.

### Catalytic studies

10 mg of degassed MOF were reacted at 30 °C with aniline (80 µL, 0.08 mmol), cyclohexene oxide (80 µL, 0.08 mmol) and *n*-dodecane (80 µL, 0.06 mmol) added as internal standard. Small aliquots of the reacting media were taken at different times (t) with the aid of a glass pipette and diluted in ethyl acetate. A control experiment was carried out exactly in the same way, but without adding any MOF to the reacting media. Aliquot composition was analyzed using a MODEL Gas Chromatograph (GC) equipped with a Agilent 199091J-413 HP-5 column and a FID detector. Injection temperature was 250 °C. The furnace thermal ramp used for analysis was 100 °C (2 min), 200 °C (70 °C/min, 1 min) and 280 °C (40 °C/min, 3.5 min). The isolated peaks were eluted at the following retention times: 4.7 min (cyclohexene oxide, C), 5.4 min (aniline), 6.7 min (n-dodecane, D) and 9.8 min (2-(phenylamino)cyclohexan-1-ol). If we define $A = A_C/A_D$, being A the area of the isolated peaks, reaction yield (Y) can be calculated as $Y = (A_{t=0} − A_{t=x})/A_{t=0}$.

## Data availability

The processed data associated to the manuscript Figures has been deposited in Zenodo under accession code 10.5281/zenodo.8282712. Crystallographic data for the structure reported in this Article have been deposited at the Cambridge Crystallographic Data Centre, under deposition numbers CCDC 2223895 (4c). Copies of the data can be obtained free of charge via https://www.ccdc.cam.ac.uk/structures/. Additional datasets generated during the current study are available from the corresponding author on reasonable request.

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

## Acknowledgements

This work was supported by the European Union Horizon 2020 Research and innovation programme (ERC-2021-COG-101043428 & grant agreement No 823717 - ESTEEM3), the Spanish Government (CEX2019-000919-M, PID2020-118117RB-I00, RTI2018-098568-A-I00 & EUR2021-121999), the Spanish National Research Council, CSIC, (COOPA20470), the Generalitat Valenciana (PROMETEU/2021/054, MFA/2022/026 & SEJIGENT/2021/059), and the regional government of Aragon (DGA E13_20R). S.T. and A.M. thanks the Spanish government for their Ramón y Cajal contracts (RYC-2016-198 & RYC2018-024561-I). N.M.P. thanks La Caixa Foundation for a Postdoctoral Junior Leader-Retaining Fellowship (ID 100010434 and fellowship code LCF/BQ/PR20/11770014). J.C.-G. thanks the Generalitat Valenciana for his APOSTD contract (CIAPOS/2021/272) and A.R.-G., forher PhD contract (ACIF/2020/090). Z.D. acknowledges CℏEM, School of Physical Science and Technology, ShanghaiTech University (#EM02161943). We also thank the University of Valencia (NANBIOSIS) and the University of Zaragoza (National Facility ELECMI ICTS node "Laboratorio de Microscopías Avanzadas") for research facilities.

## Author contributions

Conceptualization, S.T. and C.M.-G.; Methodology, S.T.; Formal analysis, S.T.; Investigation, S.T., S.M.-G., A.R.-G., E.G.-O., J.C.-G., Z.D., A.M., N.A.-B., N.M.P., and C.M.-G.; Writing—Original Draft, S.T., A.M. and C.M.-G.; Writing—Review & Editing, S.T., S.M.-G., A.R.-G., E.G.-O., Z.D., A.M., N.A.-B., N.M.P., and C.M.-G.

## Competing interests

The authors declare no competing interests.
