## [Peer Review File · Nature Communications]

Synthetic control of correlated disorder in UiO-66 frameworksEditorial Note: This manuscript has been previously reviewed at another journal that is not operating a transparent peer review scheme. This document only contains reviewer comments and rebuttal letters for versions considered at *Nature Communications*. Mentions of the other journal have been redacted.

REVIEWER COMMENTS

Reviewer #1 (Remarks to the Author):

The authors have done an excellent job of responding to all criticisms raised [redacted]. I remain of the view that this manuscript reports high-quality experimental study demonstrating elegant control over the density and distribution of vacancies in UiO-66. It succeeds in pulling together a range of experimental techniques — spanning diffraction, TEM, gas uptake, and catalytic performance measurements — to provide new insight into defect concentration control and defect/property relationships in this important material. Given the special place occupied by UiO-66 in the MOF canon, and the importance of its defect structures in the rapidly-developing field of defect-engineered materials, I am confident that this work is of the requisite scope and quality for publication in *Nature Communications*. I recommend acceptance in its current form.

Reviewer #2 (Remarks to the Author):

The authors report a systematic study of UiO-66(Hf) synthesis as a function of ligand to metal, modulator to ligand, and temperature for formic acid modulated solvothermal synthesis in a 25 mL reactor. The authors analyze the resulting products using a range of techniques including powder X-ray diffraction, thermogravimetric analysis-mass spectrometry, gas sorption measurements, and scanning transmission electron microscopy (STEM). The authors then transfer the approach to UiO-66(Zr) and show that increasing missing metal cluster (MC) defects improves catalysis in epoxide ring-opening up to a certain defect concentration and then further defect incorporation leads to reduced activity. This variation is attributed to structural collapse associated with high defect density.

There is significant and widespread interest in defects in metal-organic frameworks, particularly in the UiO-66 family. Synthetic parameters remain under-studied in the field, particularly how their variation changes the structure and properties systematically. The manuscript accordingly is timely and presents results of interest and importance in detailing synthesis-structure-function relationships. There are a number of areas where the claims and statements appear overstated or would benefit from clarification or improved contextualisation. A revised version may be publishable in *Nature Communications*.

I have included a series of comments on the main text and supporting information (SI) below:

Main text

- Abstract: The authors write in the both the abstract and the conclusion about correlated domains 'whose relative sizes are fixed by the linker concentration'. However, from the available STEM data, this appears to be based on the observation of a relatively small domain in Fig 6c and the suggestion of visually undetectable (but present in FFT processing) of reo phases in Fig. 6b. I cannot reconcile these data with a statement about 'relative sizes' being 'fixed'. I would suggest these claims need to be moderated or additional data need to be presented to support the statements about relative size.

- p. 3: I find the attribution and history of experimental observations somewhat imprecise in this set of sentences. Ref. 21 is from 2012 and so appears somewhat confusingly after a 2014 turning point. Ref. 22 also very clearly notes that Shearer et al are not the originators of the idea for the secondary crystalline phase: "A more satisfying assignment is provided by Cliffe and Goodwin, (7b)..." (Ref. 22). The citation in Ref. 22 is: 7 (b) Cliffe, M.; Goodwin, A. Presented at EUROMAT-2013 European Congress and Exhibition on Advanced Materials and Processes, Sevilla, 2013. This original reference appears to be a conference presentation on a similar topic prior to publication of Ref. 23 by some of the same authors. Moreover, the samples, as far as I can tell, reported in Ref. 22 are for UiO-66(Zr) not UiO-66(Hf). While these might appear to be relatively minor details, representing the scientific record precisely is in fact of substantial importance. In this paragraph, it

may be useful to clarify that the phenomenon appears across UiO-66(M) materials (Note Ref. 24 is applied to UiO-66(Zr) and Ref. 25 is applied to UiO-66(Hf)).

- p. 3: The authors write that the "implications that controlling periodic disorder..." are "reliant on the establishment of synthetic routes... for directing the generation and growth of correlated defect domains." To what extent does this already exist in the literature? Ref. 23 and Ref. 25 appear to highlight a route to control. Is the purpose of this work to establish a more general or comprehensive (systematic) exploration of the parameter space? I suppose my point is that there are reports of control over defect formation in the literature already. In what ways does the systematic exploration "gain control" in a new way?

- p. 3.: Could the meaning of crystallographic purity be explained for the wide readership of Nature Communications?

- p. 3 and throughout: To my mind, the combined analysis of ML and MC vacancies is a key feature of the present work. Ref. 24 touches on both types of defects, but does not examine the interplay extensively as a function of synthetic conditions, as I understand it. However, in the present paper, there appears to be an assumption that the ML vacancies are randomly distributed, but simultaneously linked to MC cluster domain formation. To what extent is it possible to distinguish (given available data in the present work) whether the ML vacancies are correlated leading to correlated MC+ML domains or that the ML vacancies appear randomly first followed by correlated MC domain formation? What drives the correlation of the MC domains if there is no correlation of ML in space?

- Figure 1: I find this representation to be somewhat misleading visually as the gray triangle conveys a sense of unexplored parameter space (with a significant area in the visual summary) when in fact there are significantly more reported synthetic points within this region. I note that Ref. 23 appears to report 15 different conditions (across Fig. 2f and Fig. 2g), but '23' is only marked for 3 conditions. I also note that Ref. 25 appears to report 4 different L/M ratios but only 2 conditions appear to be captured in the diagram. I think it would be important to convey the density of sampling across the gray triangle from existing reports to fairly contextualize the results in the present work. If the plot as constructed does not lend itself to plotting all of the synthetic parameters already reported (I would hope it could be made so), then a further alternative might be to draw in dashed lines where there are additional ranges explored, with some indication of the number of syntheses across those parameter space vectors. I am also not convinced the nonlinear y axis scale is as clear as it needs to be. I can appreciate there is an intention to show the density and variety of solvents and modulators at 1.0, 1.4 and 2.0 for the L/M ratio. However, the difference in scale between the 1.0 and 2.0 and the 1.4 tick marks is visually misleading. I think the presentation could be improved by ensuring these 'blocks' along the y axis are of equal size and to consider proportionately scaling the placement of the tick mark blocks relative to each other to retain information on the y axis. At present, the y axis is somewhere between categorical (binned) data rather and a continuous variable. The relative sizing and spacing should at least seek to preserve the continuous variable content as much as possible, even if that requires extending the physical dimension of the y axis to improve the depiction.

- p. 6: For reproducibility, would the authors be able to include more details of the geometry of the used 'Teflon bottles'? Are these containers Teflon only? How are they sealed if so? Or are these Teflon-lined steel autoclaves or something else?

- Figure 2: Why P2/P3 rather than P1/P3? I note that in Fig. 2a the P1/P3 ratio may be very different to the P2/P3 ratio. Has this been considered?

- p. 9 and elsewhere: The term "defectivity" generally seems poorly defined as an umbrella for ML and MC defects concentrations. Is defect number or concentration what is meant?

- Figure 3: Why are open circles different sizes in panels a) and b)?

- Figure 3: For the interpolation in b) does the red (low cluster connectivity) appear purely as a function of interpolation between the points or is there data underpinning these curves? It is hard

to tell from the plot - the caption suggests that the interpolation is only from $T = 120\text{ }^{\circ}\text{C}$ data, so I am not sure I can see how the cluster connectivity between the filled and unfilled circles is anything other than extrapolation to the mid-point between the closes and open circles. While it provides an appealing visualization, it may be better to demarcate what is strictly interpolation and what is extrapolation to the edge of the space taken as interpretable. If I have misunderstood something about the presented data, it may also or additionally be useful to clarify the interpolation further in revisions to the text.

- p. 11: I think the statement "Our data also reveal a clear preference for the formation of 8-c Hf6L8 clusters..." needs to be presented much more cautiously. As I understand it, these measurements show a tendency for an average cluster connectivity of 8. This in no way contradicts the formation of 4-c and 12-c clusters or 6-c and 8-c clusters or some other distribution with a center at 8-c. The phrasing suggests that the center of the distribution is the most favored, but it is not clear that this is the case (or that the shape of the N-c distribution can be ascertained from the approach employed). I do not think this is purely pedantic or semantic as a point - notably, the authors refer to 'formation', i.e. the chemical process - and it should be made clear what the limitations on inference about the chemical process are given the facts of the observables about the system.

- p. 14: I would suggest the authors consider (and possibly note) that a unit cell description will necessarily also hide any distribution of linker connectivity.

- Figure 5 (minor point): What determines the increase in adsorption at 0.9-1 P/P0 for some (but not all) samples?

- Figure 6: Are these the only 4 images acquired? Given the weight placed on the STEM data in the abstract and throughout the text, it would make the manuscript much stronger to show (in the SI, for example) further examples of each. At the moment, it appears that there is a single example ($n = 1$) for each condition, which makes it difficult to determine whether these are representative features or examples drawn to match expectations from the XRD. The number of examples need not be large if there is a high degree of correspondence between replicates.

- Figure 6: I would encourage the authors to comment on the surface termination features visible in Figure 6d and S19.

- Figure 6: Based on the SI text, are these images filtered as presented? It would be important to state filtering applied in the caption text for transparency.

- p. 17: What do the authors mean exactly by Fourier Diffractogram? The Fast Fourier Transform applied to an image is not a diffraction pattern as it represents the spatial frequencies in the image rather than the intersection of an Ewald sphere with a 3D Fourier transform of the sample (or a camera placed at the crossover plane after full wave propagation of the incident wavefront through the sample). Moreover, a Fourier transform of an image is a complex function - is the modulus or the modulus squared of this function plotted in the 'FD'? I would suggest not using 'diffractogram' for image processing (including FFT-based processing) where there is no diffraction pattern formed from the crystal. There are similarities between an FFT of an image of a crystal and a diffraction pattern, but there are also key differences - for example, the FFT will necessarily always exhibit perfect inversion symmetry as a Fourier transform of a two-dimensional real-valued function. However, a true diffraction pattern (even taking a real-valued three-dimensional potential) will not capture the Friedel pair inversion symmetry in the three-dimensional reciprocal lattice in all cases (see for example the formation of Laue circles without two-dimensional inversion symmetry for slightly off-axis electron diffraction data). Consequently, an ADF-STEM image will have greater symmetry than the true diffraction pattern for the same crystal (e.g., when slightly off-axis and atomic columns remain visible in the image).

- p. 17: Is the ABF imaging simultaneous or sequential? I think this detail would be important to include in the main text for clarity. Is the ABF imaging from the same particles or regions shown in the ADF-STEM or from further particles/regions? Are there ABF overview images that could be included in the SI?

- p. 18: The conclusion that 'lower spatial resolution achieved' is 'associated to the lower stability under the e-beam' would appear to be somewhat overstated. Have the authors quantitatively analyzed the relative stability? A reduced number of spots detected in an FFT could have many origins. For example, the field of view used for the FFT could be different, the thickness of the sample could be different, the defocus at the sample could be different, the aberrations in the STEM probe may not be the same, the sample may be less stable on the grid or the stage may not be as settled. The fluence history of the sample would also need to be documented as any pre-exposure for positioning the sample in the field of view could also contribute to beam damage that does not reflect the relative stability. A systematic selected area diffraction or image/FFT based fluence series to extract critical fluence values for different samples could provide some further support for this claim. Alternatively or in addition, a more representative set of images could provide a little bit more evidence that these observations are reproducible across particles. Importantly, it is not possible to see how consistent the STEM observations are across or within a single sample. Do all the crystals in the least defective sample show the same quality of image and FFT? I find it somewhat confusing why the Fourier transformed data are presented at different scales as it also makes it impossible to see the authors reported claim that fewer spots are visible in f) relative to e).

- p. 19: I am unable to identify evidence for the assertion that 'It is important to stress out that the materials with missing units, linkers and specially clusters, present much lower e-beam stability...' in the manuscript in its current form. While this statement may be true, and would not be entirely surprising, the wealth of other reasons for why images and FFTs exhibit reduced resolution and fewer spatial frequency spots do not appear to be discountable.

- General point (not a factor in overall manuscript evaluation): While not a basis for recommendation on publication, I would encourage the authors to correct some minor spelling or grammatical errors (e.g. 'proof' vs 'prove', 'refion', 'as experimental platform' vs 'as an experimental platform', and similar).

Supporting information:

- Section 1.4: How was "assuring a spatial resolution of 0.8 Å" assessed exactly? How long (and under what conditions) does an alignment provide this resolution on this instrument?

- Section 1.4: What fluence was used ("low electron dose")? Does the upsampling algorithm imply the images presented in the manuscript are filtered? This should be stated in the captions if that is the case. If the raw data is available, it would be good to include either in the SI or in a data repository alongside any filtered/processed data.

- Section 1.4: What are the inner and out collection angles for the ADF and ABF imaging conditions? What were the defocus values selected for image simulations and how were these determined (were a series of defocus values and thickness values explored and compared to determine the optimal match)? Is Cs the only aberration included? It is well known that, particularly low order aberrations, may drift over time (e.g. two-fold astigmatism and/or coma). How have these aberrations been handled in the simulations?

- Section 1.4: What do the authors mean by "similarity to the experimental observation"? Does this mean that a judgment was made on the experimental images directly about the sample thickness or that the authors compared a thickness series of simulations to match the experimental data? If the latter, what thickness range was explored and how finely incremented were these simulations across different thicknesses?

- Figure S2: Fig S2c shows peaks that appear to be another phase for L/M 0.2-0.6. Have these been indexed and/or are they discussed in the text?

- Figure S9: The use of color bars covering different ranges of Mod/L ratios is visually confusing. It would be better to plot the colors consistently across all samples to support inter-comparison.

- Figure S19: It appears this is in fact the same particle shown in Fig. 6c but with some rotation applied. This information should be included as a clear statement in the caption here. The "T = 120 °C, L/M = 0.4 sample" could be read as being a particle from the same batch rather than the same particle. The rotation between the image here and Fig. 6c makes it less obvious that this is the identical particle shown across the two figures.

RESPONSE TO REVIEWERS' COMMENTS

We have made a number of changes, additions and revisions to the manuscript addressing the suggestions made by the reviewers and correcting some additional errors that we have detected during the review process. A point-by point response to their comments can be found below along with a detailed list of additional changes.

We would like to express our sincere thanks to the reviewers for their positive assessment and constructive criticism. We hope we have successfully addressed their concerns that have certainly helped strengthening the manuscript.

Reviewer 1

Comments

The authors have done an excellent job of responding to all criticisms raised [redacted]. I remain of the view that this manuscript reports high-quality experimental study demonstrating elegant control over the density and distribution of vacancies in UiO-66. It succeeds in pulling together a range of experimental techniques — spanning diffraction, TEM, gas uptake, and catalytic performance measurements — to provide new insight into defect concentration control and defect/property relationships in this important material. Given the special place occupied by UiO-66 in the MOF canon, and the importance of its defect structures in the rapidly-developing field of defect-engineered materials, I am confident that this work is of the requisite scope and quality for publication in Nature Communications. I recommend acceptance in its current form.

Response

We thank the reviewer for recognizing the effort made to address all his/her comments and for recognizing the quality and scientific appeal of our manuscript.

Reviewer 2

Comments

The authors report a systematic study of UiO-66(Hf) synthesis as a function of ligand to metal, modulator to ligand, and temperature for formic acid modulated solvothermal synthesis in a 25 mL reactor. The authors analyze the resulting products using a range of techniques including powder X-ray diffraction, thermogravimetric analysis-mass spectrometry, gas sorption measurements, and scanning transmission electron microscopy (STEM). The authors then transfer the approach to UiO-66(Zr) and show that increasing missing metal cluster (MC) defects improves catalysis in epoxide ring-opening up to a certain defect concentration and then further defect incorporation leads to reduced activity. This variation is attributed to structural collapse associated with high defect density.

There is significant and widespread interest in defects in metal–organic frameworks, particularly in the UiO-66 family. Synthetic parameters remain under-studied in the field, particularly how their variation changes the structure and properties systematically. The manuscript accordingly is timely and presents results of interest and importance in detailing synthesis-structure-function relationships. There are a number of areas where the claims and statements appear overstated or would benefit from clarification or improved contextualisation. A revised version may be publishable in Nature Communications.

I have included a series of comments on the main text and supporting information (SI) below:

Main text:

Critique 1

- Abstract: The authors write in the both the abstract and the conclusion about correlated domains 'whose relative sizes are fixed by the linker concentration'. However, from the available STEM data, this appears to be based on the observation of a relatively small domain in Fig 6c and the suggestion of visually undetectable (but present in FFT processing) of reo phases in Fig. 6b. I cannot reconcile these data with a statement about 'relative sizes' being 'fixed'. I would suggest these claims need to be moderated or additional data need to be presented to support the statements about relative size.

Response to critique 1

We use “fixed” as a synonym of “predetermined”, “set” or “stablished” and draw this conclusion from different pieces of evidence, and not only from STEM. For instance, the Rietveld refinement of several samples reveal an increase of the fraction of **reo** phase with decreasing linker concentration (Figure 4a) up to a value close to 90%. As the size of the crystals does not change with ligand concentration (Figure S15),

net reduction of **fcu** fractions in the crystals will necessarily involve bigger **reo** fractions or domains.

Nevertheless, based on the STEM study and also confirmed by FFT data, figure 6c corroborates the existence of both domains within the same particle marked by different colors, while Figure 6 a,b and Figure 6d shows either **fcu** or **reo** domains. Also supporting our original hypothesis on how domain sizes are dependent on the concentration of linker used in the synthesis.

Critique 2

- p. 3: *I find the attribution and history of experimental observations somewhat imprecise in this set of sentences.*

Ref. 21 is from 2012 and so appears somewhat confusingly after a 2014 turning point.

Ref. 22 also very clearly notes that Shearer et al are not the originators of the idea for the secondary crystalline phase: "A more satisfying assignment is provided by Cliffe and Goodwin, (7b)..." (Ref. 22).

The citation in Ref. 22 is: 7 (b) Cliffe, M.; Goodwin, A. Presented at EUROMAT-2013 European Congress and Exhibition on Advanced Materials and Processes, Sevilla, 2013. This original reference appears to be a conference presentation on a similar topic prior to publication of Ref. 23 by some of the same authors.

Moreover, the samples, as far as I can tell, reported in Ref. 22 are for UiO-66(Zr) not UiO-66(Hf). While these might appear to be relatively minor details, representing the scientific record precisely is in fact of substantial importance. In this paragraph, it may be useful to clarify that the phenomenon appears across UiO-66(M) materials (Note Ref. 24 is applied to UiO-66(Zr) and Ref. 25 is applied to UiO-66(Hf)).

Response to critique 2

Thank you very much for clarifying this point regarding the chronology of the experimental evidence that has been building the concept of defect engineering in this family of MOFs. We fully agree with this view and in revising the paper to try to fix the problem we realised that there was a clear error in this paragraph. Our previous revision included a paragraph that completely misrepresented the actual chronology of events. Our idea was to highlight that the turning point arrived with the attribution of superlattice peaks arising from ordered missing clusters.

With respect to the introduction of the communication by Cliffe and Goodwin presented at the EUROMAT-2013 held in Sevilla, we have not been able to find a concrete document that would allow us to analyze it in detail. Unfortunately, none of the co-authors were present at the conference and we cannot attest to this fact. In any case, we have full confidence in the reviewer's opinion and have decided to include a reference to this communication.

We agree with the referee that making the difference between UiO-66(Zr/Hf) is a relative minor detail. Even more so for this manuscript as we produce both, defective UiO-66(Zr) and UiO-66(Hf) by following the same synthetic approach, with comparable results and using the same experimental procedures.

Bearing in mind that the main objective of the introduction to our work is to give the reader a general idea of how the area has evolved from the empirical identification of the presence of defects, their structural origin, as well as the possibility of using synthetic tools to learn how to control their generation and distribution in the solid, we consider that making this distinction for the examples described in the literature could hinder the understanding of the general context. We respectfully consider that making this distinction is not necessary in our case.

To account for all this, we have made the following changes:

Page 3 from:

[...] While it was initially assumed that defects existed in the form of randomly distributed missing linkers (ML), a turning point arrived in 2014 when weak and broad superlattice powder X-ray diffraction peaks in Hf-UiO-66,[21,22] were unambiguously attributed to a secondary crystalline phase having practically the same unit cell size but lower symmetry. Additional diffraction lines were unambiguously attributed to a secondary crystalline phase having practically the same unit cell size but lower symmetry.[23]

To:

[...] While it was initially assumed that defects existed in the form of randomly distributed missing linkers (ML), a turning point arrived in 2014 when the presence of weak and broad superlattice powder X-ray diffraction peaks in UiO-66,[21,22, Additional reference] were unambiguously attributed to a secondary crystalline phase based on metal cluster vacancies having practically the same unit cell size but lower symmetry.[23]

[Additional Reference]

Cliffe, M.; Goodwin, A. Presented at EUROMAT-2013 European Congress and Exhibition on Advanced Materials and Processes, Sevilla, 2013. b) Cliffe, M. J.; Wan, W.; Zou, X.; Chater, P. A.; Kleppe, A. K.; Tucker, M. G.; Wilhelm, H.; Funnell, N. P.; Coudert, F.-X.; Goodwin, A. L. Correlated Defect NanoRegions in a Metal–Organic Framework. *Nature Communications* **2014**, 5, 4176–4176.

Critique 3

- p. 3: The authors write that the “implications that controlling periodic disorder...” are “reliant on the establishment of synthetic routes... for directing the generation and growth of correlated defect domains.” To what extent does this already exist in the literature? Ref. 23 and Ref. 25 appear to highlight a route to control. Is the purpose of this work to establish a more general or comprehensive (systematic) exploration of

the parameter space? I suppose my point is that there are reports of control over defect formation in the literature already. In what ways does the systematic exploration "gain control" in a new way?

Response to critique 3

We thank the reviewer for his/her insights on the matter, but we do not intend to start a sterile discussion on the extent to which references 23 and 25 establish general synthetic routes for controlling the generation and growth of correlated defect domains in UiO-66. These works are very valuable and are accordingly cited in the manuscript, but they do not deploy the systematic exploration of chemical variables that are central to our manuscript. This systematic approach has enabled to capture the region of the chemical space in which pure **reO** UiO-66 phases can be reproducibly isolated for the first time. This was one of our main objectives, to provide the community with synthetic tools to generate pure UiO-66 defective phases in a controlled way, instead of working with phase mixtures and the difficulties that this entails for the rationalisation of their physical properties and catalytic activity.

Critique 4

- p. 3.: Could the meaning of crystallographic purity be explained for the wide readership of Nature Communications?

Response to critique 4

Based on IUPACs Goldbook [Response Reference] a "phase" is an entity of a material system which is uniform in chemical composition and physical state. A "phase" will be "crystalline" if there is three-dimensional order on the level of atomic dimensions. As result, the structure of the crystalline phase can be formed by regularly repeating in three dimensions a material portion contained in a parallelepiped, the so-called "unit cell". The positions and intensities of the peaks in a powder pattern are determined by the arrangement of atoms in the unit cell. Summing up, in our opinion a crystallographic pure phase can be defined as the one in which the position and intensities of all the peaks in a powder diffraction pattern can be assigned to a single crystalline phase.

[Response Reference] IUPAC Gold Book, <https://goldbook.iupac.org/>

To comply with the referee's request, we have made the following modification:

Page S4 from:

[...] LeBail method in all the samples to confirm the space group and phase purity.

To:

LeBail method in all the samples to confirm the space group and phase purity. A pure crystallographic phase is defined as a phase in which the position and intensities of

all the peaks in a powder diffraction pattern can be assigned to a single crystalline phase.

Critique 5

- p. 3 and throughout: To my mind, the combined analysis of ML and MC vacancies is a key feature of the present work. Ref. 24 touches on both types of defects, but does not examine the interplay extensively as a function of synthetic conditions, as I understand it. However, in the present paper, there appears to be an assumption that the ML vacancies are randomly distributed, but simultaneously linked to MC cluster domain formation. To what extent is it possible to distinguish (given available data in the present work) whether the ML vacancies are correlated leading to correlated MC+ML domains or that the ML vacancies appear randomly first followed by correlated MC domain formation? What drives the correlation of the MC domains if there is no correlation of ML in space?

Response to critique 5

That is an excellent question. There are several issues here, the first and most straightforward is that the scattering factor of Hafnium is much higher than that of carbon. To try to illustrate this we attach the PXRD of (**fcu**-UiO-66 (4 clusters 24 linkers), **reo**-UiO-66 (3 clusters 12 linkers) and **bcu**-UiO-66 (4 clusters 16 linkers):

Even if ML are ordered (**bcu** case) it will not be easy to detect this correlation using PXRD.

On the other hand, there are multiple possibilities of removing 8 linkers from the **fcu** unit cell, all of them leading to an average cell cluster connectivity of 8 and a cell with lower symmetry, but only one will lead to the high symmetry **bcu** cell. Contrary to that, there is only one possibility configuration that will results in a MC (all the twelve missing linkers removed from the same cluster, cluster connectivity 0). Summing up, if we remove a high number of linkers from the **fcu** unit cell is statistically more probable to end up with ordered missing clusters (**reo**) than linkers (**bcu**). This is why, as we keep removing linkers, the average cluster connectivity decreases, to a point that triggers the collapse of some domains in the framework, but the chances of getting 0 connected clusters one next to the other increases, leading to missing cluster correlations (Figure 3c and Figure 4a in the manuscript).

Critique 6

- *Figure 1: I find this representation to be somewhat misleading visually as the gray triangle conveys a sense of unexplored parameter space (with a significant area in the visual summary) when in fact there are significantly more reported synthetic points within this region. I note that Ref. 23 appears to report 15 different conditions (across Fig. 2f and Fig. 2g), but '23' is only marked for 3 conditions. I also note that Ref. 25 appears to report 4 different L/M ratios but only 2 conditions appear to be captured in the diagram. I think it would be important to convey the density of sampling across the gray triangle from existing reports to fairly contextualize the results in the present work. If the plot as constructed does not lend itself to plotting all of the synthetic parameters already reported (I would hope it could be made so), then a further alternative might be to draw in dashed lines where there are additional ranges explored, with some indication of the number of syntheses across those parameter space vectors. I am also not convinced the nonlinear y axis scale is as clear as it needs to be. I can appreciate there is an intention to show the density and variety of solvents and modulators at 1.0, 1.4 and 2.0 for the L/M ratio. However, the difference in scale between the 1.0 and 2.0 and the 1.4 tick marks is visually misleading. I think the presentation could be improved by ensuring these 'blocks' along the y axis are of equal size and to consider proportionately scaling the placement of the tick mark blocks relative to each other to retain information on the y axis. At present, the y axis is somewhere between categorical (binned) data rather and a continuous variable. The relative sizing and spacing should at least seek to preserve the continuous variable content as much as possible, even if that requires extending the physical dimension of the y axis to improve the depiction.*

Response to critique 6

Figure 1 has been reworked to include all the reported synthetic conditions in the cited articles. As suggested by the referee we added lines as visual aids to track the

synthetic conditions and made 'blocks' along the y axis of equal size. In our opinion, and as far as we understand the “continuous variable content”, was already preserved. To try to stress this point we added some additional tick to the y-axis to make this effect more apparent.

In what regards Table 1, it was meant to aid the reader to have a more precise view over experimental conditions by including conditions such as reaction time, metal salts and linker used. In this revised version of the manuscript, Graph 1 contains 57 points, a comprehensive version of Table 1 will span through two pages and will contain basically no additional information compared to the reduced version (Mod/L, L/M and Temperature are already reported in Figure 1). For that reason, we decided to keep a reduced version of Table 1.

Figure 1 and caption from:

Figure 1: Summary of the experimental conditions used by different authors to produce UiO-66 structures with missing cluster defects. Data selection is based on the bibliographic survey by Feng et al.[29] When multiple ratios were reported by the same authors (e.g. 23 and 25), we have only represented the highest and lowest ratios. Please note that, for clarity, axis do not follow a linear scale. y and x axis display respectively, the linker/metal (L/M) and modulator/linker (Mod/L) ratios used in all works selected. Modulators: FA (formic acid), TFA (trifluoroacetic acid), AA (acetic acid) and BA (benzoic acid). Solv refers to the solvent used during the synthesis. Linkers used were either benzene-1,4-dicarboxylic or 2-aminobenzene-1,4-dicarboxylic acids and metal salts ZrCl₄, ZrOCl₂ or HfCl₄. The chemical space explored in this work is delimited with a grey triangle. We refer the reader to Table S1 for a detailed description of the experimental conditions, corresponding references are signaled with small numbers. For reference 39 we have considered H₂O = Mod.

To:

Figure 1: Summary of the experimental conditions used by different authors to produce UiO-66 structures with missing cluster defects. Data selection is based on the bibliographic survey by Feng et al.[29] Please note that, for clarity, axis do not follow a linear scale. y and x axis display respectively, the linker/metal (L/M) and modulator/linker (Mod/L) ratios used in all works selected. Modulators: FA (formic acid), TFA (trifluoroacetic acid), AA (acetic acid) and BA (benzoic acid). Solv refers to the solvent used during the synthesis. Linkers used were either benzene-1,4-dicarboxylic or 2-aminobenzene-1,4-dicarboxylic acids and metal salts $ZrCl_4$, $ZrOCl_2$ or $HfCl_4$. The chemical space explored in this work is delimited with a grey triangle. We refer the reader to Table S1 for more detailed description of the experimental conditions, corresponding references are signaled with small numbers.

Table S1 caption from:

Summary of the conditions used by different authors to obtain UiO-66 structures with missing cluster type of defects

To:

Summary of the conditions used by different authors to obtain UiO-66 structures with missing cluster type of defects. When multiple ratios were reported by the same authors, we have only summarized the highest and lowest ratios.

Critique 7

- p. 6: For reproducibility, would the authors be able to include more details of the geometry of the used 'Teflon bottles'? Are these containers Teflon only? How are they

sealed if so? Or are these Teflon-lined steel autoclaves or something else?

Response to critique 7

To try to help others with the reproducibility of our experiments we have added the following modification:

Page S3 from:

[...] and BDC into a 25 mL teflon bottles[...]

To:

[...] and BDC into VWR 89026-060 PTFE (height, body diameter, neck diameter = 61, 33, 19 mm) bottles[...]

VWR include pictures of the bottle in its online catalogue that are easily accessible using the catalogue number 89026-060 provided.

Critique 8

- Figure 2: Why P2/P3 rather than P1/P3? I note that in Fig. 2a the P1/P3 ratio may be very different to the P2/P3 ratio. Has this been considered?

Response to critique 8

Thanks for pointing this out. Yes, it has been considered. We checked that P2/P3 and P1/P3 follow roughly the same trend with linker concentration. It is this trend the one we used to draw qualitative conclusions, while Rietveld refinement was used (all the peaks are considered) for qualitative analysis.

From a practical point of view, at very low *reo* fractions P1 tends to be wider than P2, background contribution to the overall diffraction pattern is also more intense at low angles. So, at high linker concentrations in general I_{P1} , if observable, is more difficult to measure than I_{P2} . That is why we originally chose P2.

Critique 9

- p. 9 and elsewhere: The term “defectivity” generally seems poorly defined as an umbrella for ML and MC defects concentrations. Is defect number or concentration what is meant?

Response to critique 9

The referee is right. As we explained before, it is not easy to make the difference between ML and MC. ML formation takes place continuously and finally leads to MC formation and spatial correlation. Defectivity in the wider sense means stoichiometry deviating from that of *fcu*-UiO-66. In our opinion, for this case the number of defects and their relative concentration are two sides of the same coin and can be considered equivalent.

Critique 10

- Figure 3: Why are open circles different sizes in panels a) and b)?

Response to critique 10

Thanks for pointing this out. We missed this detail. The figure has been changed accordingly to make the size of both open circle sets equivalent. We have also identified an error in the caption of the figure that has been now modified.

Figure 3 from:

Figure 3: Defined area for chemical control of defects and proposed mechanism for generation and growth of correlated domains in Hf-UiO-66. Compositional plots interpolated from Series A, B and C data at $T = 120^{\circ}\text{C}$. a) Correlation of superlattice diffraction and b) Cluster connectivity as a function of the L/M and Mod/L ratios. Samples in the grey area correspond to the formation of secondary crystallographic phases (open circle symbol) or the inability to form a UiO-66-like solid (cross symbol). c) Proposed structural description for the different stages in the formation and growth of reo domains starting from defect-free fcu topology and accounting for the variations in superlattice diffraction and cluster connectivity observed for variable L/M ratio.

To:

Figure 3: Defined area for chemical control of defects and proposed mechanism for generation and growth of correlated domains in Hf-UiO-66. Compositional plots interpolated from Series A, B and C data at $T = 120^\circ\text{C}$. The colour-z-axis used respects the numerical value of the experimental point used in each case. a) Correlation of superlattice diffraction and b) Cluster connectivity as a function of the L/M and Mod/L ratios. Samples in the grey area have not been considered for analysis. Open circle and cross symbols correspond to the formation of secondary phases and the inability to form a UiO-66-like solids, respectively. c) Proposed structural description for the different stages in the formation and growth of reo domains starting from defect-free fcu topology and accounting for the variations in superlattice diffraction and cluster connectivity observed for variable L/M ratio.

Critique 11

- Figure 3: For the interpolation in b) does the red (low cluster connectivity) appear purely as a function of interpolation between the points or is there data underpinning these curves? It is hard to tell from the plot - the caption suggests that the interpolation is only from $T = 120^\circ\text{C}$ data, so I am not sure I can see how the cluster connectivity between the filled and unfilled circles is anything other than extrapolation to the midpoint between the closes and open circles. While it provides an appealing visualization, it may be better to demarcate what is strictly interpolation and what is extrapolation to the edge of the space taken as interpretable. If I have misunderstood

something about the presented data, it may also or additionally be useful to clarify the interpolation further in revisions to the text.

Response to critique 11

The referee is right, as stated on the caption of Figure 3: the interpolation corresponds to compositional plots interpolated from Series A, B and C data exclusively at $T = 120^{\circ}\text{C}$. As far as the interpolation is concerned, the referee's doubts are reasonable and that is why we decided to include together with the 2D map resulting from the interpolation also the real points that were used for it. The colour-z-axis used respects the numerical value of the experimental point used in each case (no interpolation). For clarity, we also identify the points that correspond to a phase mixture or with the impossibility to form identifiable phases as open circle or cross symbols respectively. All data used in the graphical representation in Figure 3 correspond directly to the separately analysed data sets collected in both the main text and the Supplementary section.

In any case, the main objective of this representation is to try to convey to the reader a representative picture of the variations in the presence of characteristic supercell diffraction and changes in the average cluster connectivity index in the compositional space studied. From the referee's comments we understand that he/she also appreciates the visual appeal of this type of representation to help simplify the density of data available. For this reason, we have decided to respect the figure and add an explanatory note to the legend:

Page 11, caption Figure 3 from:

[...] Compositional plots interpolated from Series A, B and C data at $T = 120^{\circ}\text{C}$. [...]

To:

[...] Compositional plots interpolated from Series A, B and C data at $T = 120^{\circ}\text{C}$. The colour-z-axis used respects the numerical value of the experimental point used in each case. [...]

Critique 12

- p. 11: I think the statement "Our data also reveal a clear preference for the formation of 8-c Hf6L8 clusters..." needs to be presented much more cautiously. As I understand it, these measurements show a tendency for an average cluster connectivity of 8. This in no way contradicts the formation of 4-c and 12-c clusters or 6-c and 8-c clusters or some other distribution with a center at 8-c. The phrasing suggests that the center of the distribution is the most favored, but it is not clear that this is the case (or that the shape of the N-c distribution can be ascertained from the approach employed). I do not think this is purely pedantic or semantic as a point - notably, the authors refer to 'formation', i.e. the chemical process - and it should be

made clear what the limitations on inference about the chemical process are given the facts of the observables about the system.

Response to critique 12

We agree with the referee on the importance of being as cautious as possible when interpreting experimental results. Pointing to the thermodynamic preference for the formation of 8-c Hf6L8 clusters was only intended to explain why our compositional analyses always reveal connectivity indices equal or lower than 8 when the synthesis temperature is equal or lower than 120 °C. We agree on the possibility that this average connectivity index could also be the result of the concurrence of clusters with connectivity higher and lower than 8, whose potential formation cannot be excluded, resulting in a distribution of connectivity centered on 8 as the average value. Unfortunately, we do not have, and do not know how to obtain, experimental data that would allow us to identify locally the connectivity indices in such a compositionally complex sample. Thermogravimetric analysis is an average technique, and the result only reflects the prevalence of connectivity indices 8 in the samples analyzed. Whether the origin of this preference is thermodynamic or not is something we are not able to establish unequivocally.

This is why we have decided to modify the text so as not to incur inaccuracies to a thermodynamic preference in the formation of 8-c clusters and simply point to their prevalence according to our experimental data.

Page 8 from:

To confirm this point, we used TGA to correlate synthetic variations with changes to the cluster connectivity.

To:

To confirm this point, we used TGA to correlate synthetic variations with changes to the average cluster connectivity, which from this point on we will refer to as cluster connectivity for brevity's sake.

Page 11 from:

Our data also reveal a clear preference for the formation of 8-c Hf6L8 clusters regardless the experimental conditions used.

To:

Our data also reveal the prevalence of clusters with connectivity 8 regardless the experimental conditions used.

Critique 13

- p. 14: I would suggest the authors consider (and possibly note) that a unit cell description will necessarily also hide any distribution of linker connectivity.

Response to critique 13

We agree with the reviewer. The text has been modified accordingly:

Page 14 from:

[...] The refinement of linker occupancies in the structure is also consistent with the preference for 8-c Hf₆L₈ clusters at T = 120°C described above.

To:

[...] The refinement of linker occupancies in the structure is also consistent with the prevalence of the clusters with connectivity 8 at T = 120°C described above.

Critique 14

- *Figure 5 (minor point): What determines the increase in adsorption at 0.9-1 P/P0 for some (but not all) samples?*

Response to critique 14

The presence of this adsorption tail at these relative pressures is normally associated with samples composed of small-sized crystals. The presence of submicrometric crystals in our samples, according to the SEM images shown in Figure S15, will generate a certain intergrain mesoporosity above P/P0 0.9 as a result of the packing in the solid of very small particle sizes. This behaviour is quite common and more acute for those samples in which the crystal size is comparatively smaller. In our case, the 1.5 (T = 180 °C) and 0.1 (T = 120 °C) samples displayed bigger crystal sizes which reduces the impact of this phenomenon as observed in Figure 5a.

This behaviour is quite common and we did not consider it necessary to provide further information on this issue.

Critique 15

- *Figure 6: Are these the only 4 images acquired? Given the weight placed on the STEM data in the abstract and throughout the text, it would make the manuscript much stronger to show (in the SI, for example) further examples of each. At the moment, it appears that there is a single example (n = 1) for each condition, which makes it difficult to determine whether these are representative features or examples drawn to match expectations from the XRD. The number of examples need not be large if there is a high degree of correspondence between replicates.*

Response to critique 15

In this revised version of the manuscript, we have included the images already provided in our previous revision of the as additional supplementary images.

Page 17 from:

and defective samples synthesized at 120°C for varying linker concentrations are shown in Figure 6a-h.

To:

and defective samples synthesized at 120°C for varying linker concentrations are shown in Figure 6a-h and [Additional Supporting Figures from Critique 15 and Critique 20].

Page S26:

Cs-corrected STEM images of different Hf-UiO-66 frameworks obtained. When both ***fcu*** and ***reo*** domains are present in the same particle they have been signaled with blue and red arrows, respectively. Moreover, the surface reconstruction present in some of the images has been highlighted with yellow arrows. We do not know what the exact nature of those surface reconstructions might be at the moment. Possible candidates could be related to ***hcp***, ***hns*** or ***hxl*** structures[Additional SI Reference 1] or Hafmium formate.[Additional SI Reference 2] To try to get more information on this hypothesis, we are currently trying to obtain information along different zone axes, but it is resulting very challenging due to the morphology of the particles and their preferential [110] zone axis.

Figure SX. Cs-corrected STEM analysis of a T = 180°C, L/M = 1.5 Hf-UiO-66 sample.

Figure SX. Cs-corrected STEM analysis of a $T = 120^{\circ}\text{C}$, $L/M = 1.5$ Hf-UiO-66 sample.

Figure SX. Cs-corrected STEM analysis of a $T = 120^{\circ}\text{C}$, $L/M = 0.4$ Hf-UiO-66 sample.

Figure SX. Cs-corrected STEM analysis of a $T = 120^{\circ}\text{C}$, $L/M = 0.1$ Hf-UiO-66 sample.

[Additional SI reference 1] Firth FCN, Cliffe MJ, Vulpe D, Aragones-Anglada M, Moghadam PZ, Fairen-Jimenez D, Slater B, Grey CP. *J. Mater. Chem. A*. 7, 7459–7469 (2019).

[Additional SI reference 2] Liang, W., Babarao, R., Murphy, M. J. & D’Alessandro, D. M. The first example of a zirconium-oxide based metal–organic framework constructed from monocarboxylate ligands. *Dalton Trans.* 44, 1516–1519 (2015).

Critique 16

- *Figure 6: I would encourage the authors to comment on the surface termination features visible in Figure 6d and S19.*

Response to critique 16

We are also very interested in finding out the origin of the observed crystal surface reconstruction, unfortunately we are not yet able to give an explanation at this stage. We would not like to anticipate too preliminary a hypothesis that would not respect the necessary scientific standards for this forum. The referee can be assured that we are working in this direction and hope to be able to share specific work on the surface reconstruction of such systems as soon as we have all the necessary experimental data, possible candidates could be related to *hcp*, *hns* or *hxl* structures. We added a short comment at this respect, see response to Critique 15.

Critique 17

- Figure 6: Based on the SI text, are these images filtered as presented? It would be important to state filtering applied in the caption text for transparency.

Response to critique 17

We sincerely do not believe this is necessary as it was already clarified in Page S6. However, this sentence has been modified to account also for some other critiques from the referee:

Page S6 from:

[...] All images were recorded under low electron dose by using a real-time up-sampling noise filter implemented in DigitalMicrograph Software.

To:

Data acquisition and analysis of all images was performed using the Realtime module and the HREM Filters Pro from HREM Research Inc,[S7] implemented in DigitalMicrograph Software. To minimize the beam damage, the electron dose used was maintained under $700 \text{ e}^-/\text{\AA}$.

Critique 18

- p. 17: *What do the authors mean exactly by Fourier Diffractogram? The Fast Fourier Transform applied to an image is not a diffraction pattern as it represents the spatial frequencies in the image rather than the intersection of an Ewald sphere with a 3D Fourier transform of the sample (or a camera placed at the crossover plane after full wave propagation of the incident wavefront through the sample). Moreover, a Fourier transform of an image is a complex function - is the modulus or the modulus squared of this function plotted in the 'FD'? I would suggest not using 'diffractogram' for image processing (including FFT-based processing) where there is no diffraction pattern formed from the crystal. There are similarities between an FFT of an image of a crystal and a diffraction pattern, but there are also key differences - for example, the FFT will necessarily always exhibit perfect inversion symmetry as a Fourier transform of a two-dimensional real-valued function. However, a true diffraction pattern (even taking a real-valued three-dimensional potential) will not capture the Friedel pair inversion symmetry in the three-dimensional reciprocal lattice in all cases (see for example the formation of Laue circles without two-dimensional inversion symmetry for slightly off-axis electron diffraction data). Consequently, an ADF-STEM image will have greater symmetry than the true diffraction pattern for the same crystal (e.g., when slightly off-axis and atomic columns remain visible in the image).*

Response to critique 18

Most of the authors of the paper were not so familiar to Fourier Transforms, we thank the referee for this detailed and scholarly presentation to it. Now, we are all aware that we had used the term Fourier Diffractogram (FD) referring to the Fast Fourier

Transform (FFT) and that all our Fourier images contain the amplitude and phase of the wave.

We acknowledge the reviewer for the comment that helps for a better understanding and have updated the main text and Supporting Information accordingly:

Through the entire Manuscript and Supporting Information:

The term FD has been changed to FFT.

Critique 19

- p. 17: Is the ABF imaging simultaneous or sequential? I think this detail would be important to include in the main text for clarity. Is the ABF imaging from the same particles or regions shown in the ADF-STEM or from further particles/regions? Are there ABF overview images that could be included in the SI?

Response to critique 19

Again, thank you very much for this comment as it requires further clarification on our side. We have included this description in the SI:

Page S6 from:

To reduce e-beam damage, all images were recorded under low electron dose by using a real-time up-sampling noise filter implemented in DigitalMicrograph Software.[S7] Image simulations were performed using[...]

To:

Data acquisition and analysis of all images was performed using the Realtime module and the HREM Filters Pro from HREM Research Inc,[S7] implemented in DigitalMicrograph Software. To minimize the beam damage, the electron dose used was maintained under 700 e/Å. Data collection was done using simultaneously an annular dark field detector (ADF, inner-outer angles 25-160 mrad), and an annular bright field detector (ABF, inner-outer angles 5-25 mrad). Image simulations were performed using[...]

Therefore, ADF and ABF were recorded from the same region in all cases.

Critique 20

- p. 18: The conclusion that 'lower spatial resolution achieved' is 'associated to the lower stability under the e-beam' would appear to be somewhat overstated. Have the authors quantitatively analyzed the relative stability? A reduced number of spots detected in an FFT could have many origins. For example, the field of view used for the FFT could be different, the thickness of the sample could be different, the defocus at the sample could be different, the aberrations in the STEM probe may not be the same, the sample may be less stable on the grid or the stage may not be as settled. The fluence history of the sample would also need to be documented as any pre-

exposure for positioning the sample in the field of view could also contribute to beam damage that does not reflect the relative stability. A systematic selected area diffraction or image/FFT based fluence series to extract critical fluence values for different samples could provide some further support for this claim. Alternatively or in addition, a more representative set of images could provide a little bit more evidence that these observations are reproducible across particles. Importantly, it is not possible to see how consistent the STEM observations are across or within a single sample. Do all the crystals in the least defective sample show the same quality of image and FFT? I find it somewhat confusing why the Fourier transformed data are presented at different scales as it also makes it impossible to see the authors reported claim that fewer spots are visible in f) relative to e).

Response to critique 20

For all materials analysed the experimental procedure was the same. Crystals were looked in TEM mode, and tilted under TEM mode. With a magnification of 8000, always using a dose rate of $\approx 0.1 \text{ e-}/\text{\AA}^2/\text{s}$. This aspect rules out the possibility that any interaction between the beam and the sample that may cause different behavior respect to the ebeam from sample to sample. We have not described these details in the manuscript as this same procedure has been used quite conventionally for the study of beam sensitive porous solids to the best of our knowledge. Additionally, other groups may have different approaches, nor better or worse just different, such as tilting the sample by a software assisted method.

Of course, some aspects may influence the number of spots in the FFT. We always aimed for thinnest regions near the surface. The thickness was not measured in every case, but it was observed that the number of spots in the FFT decreased when the samples increased their defectivity (lower ebeam stability); we certainly doubt that the analyzed crystals were so significantly thicker in the same way as the number of defects were increased. Field of view was the same for every case, we find this aspect redundant but if the reviewer and editor consider it appropriate, we can introduce it in the manuscript.

We always try to work under the optimum focus conditions, those where the image was clearer and sharper and therefore more spots in the FFT would be present. It is true that aberrations may slightly change with time and even the method used to measure aberrations is not so precise. We run the corrector twice right one after the other without changing any parameter and values are not the same in an interval of time of 30 seconds, not in our TEM or in any TEM that we are aware of. These variations, however, are not significant in the quality of the probe and the spatial resolution achieved is always below 1\AA (under "standard" conditions used to correct the aberrations, this is sufficient electron dose). Before analyzing every crystal, coma and astigmatism were corrected using the carbon from the grid.

Before analyzing every sample, the holder was inserted and let it stay over 40 minutes to have good settlement.

Additionally, in our response to Critique 15, we provide more images of different particles, of the sample with less defects, together with their correspondent FFTs showing more spots than for the defective ones.

Regarding the fact that Fourier transformed data is presented at different scales; we have selected this option as we believe it clearer shows the aspect we want to highlight, that is, the presence of the extra spots. We would like to point out that if we had initially kept the same data scale, the reader would have missed aspects such as those that the referee pointed out in his/her Critique 4 of the previous response. However, we also understand the reviewer's point, and have included an additional Figure in Supporting information.

Additional Supplementary Figure:

Figure SX. FFTs and simulated ED patterns corresponding to Figures 6 (a-d) represented at the same scale.

Figure 6 caption from:

From left to right large area FD and simulated ED patterns corresponding to (a-d), respectively.

To:

From left to right large area FD and simulated ED patterns corresponding to (a-d), respectively; same images without scaling can be found in [Additional Figure]

Critique 21

- p. 19: I am unable to identify evidence for the assertion that 'It is important to stress out that the materials with missing units, linkers and specially clusters, present much

lower e-beam stability..." in the manuscript in its current form. While this statement may be true, and would not be entirely surprising, the wealth of other reasons for why images and FFTs exhibit reduced resolution and fewer spatial frequency spots do not appear to be discountable.

Response to critique 21

We should remark that a very similar critique was already addressed in our previous response [redacted].

Besides the changes introduced after that round of peer reviewing, in this new version of the manuscript we have softened this statement, because as the referee points out might be other reasons contributing to that.

Page 18 from:

This lower beam stability, compared to the non-defective sample, is associated to the formation of ML[...]

To:

This lower beam stability, compared to the non-defective sample, may be associated to the formation of ML[...]

Page 19 from:

It is important to stress out that the materials with missing units, linkers and specially clusters, present much lower e-beam stability, which did not allow the visualization of the organic linkers, this was also reflected by the lower amount of diffraction spots in both the FD and ED patterns.

To:

It is important to stress out that the materials with missing units, linkers and specially clusters, present lower amount of diffraction spots in the FFT, and much lower e-beam stability, which did not allow the visualization of the organic linkers.

Now the sentence is simply reflecting an experimental fact, without evaluating its cause.

Critique 22

- General point (not a factor in overall manuscript evaluation): While not a basis for recommendation on publication, I would encourage the authors to correct some minor spelling or grammatical errors (e.g. 'proof' vs 'prove', 'refion', 'as experimental platform' vs 'as an experimental platform', and similar).

Response to critique 22

We did our best to try to correct spelling and grammatical errors. Thanks for pointing this out.

Supporting information:

Critique 23

- Section 1.4: How was “assuring a spatial resolution of 0.8 Å” assessed exactly? How long (and under what conditions) does an alignment provide this resolution on this instrument?

Response to critique 23

In the experimental section, we have described the equipment and the microscope used offers 0.8 Å, we wanted to show the equipment used for the experiments. Obviously, that resolution depends on the material used to align the corrector (and to measure the resolution); we used a gold standard sample at 300 kV, with enough signal-to-noise ratio. From the experience of several users, this alignment is stable for several weeks. Nevertheless, to assure the best conditions possible, the corrector was aligned (corroboration its stability) before the experiments.

We would like to note that this resolution was achieved by routine TEMs many years ago. To try to stress this point we have made the following change:

Page S6 from:

[...] which was aligned prior the experiments assuring a spatial resolution of 0.8 Å.

To:

[...] which was aligned prior the experiments assuring a spatial resolution on a gold standard sample below 0.8 Å.

Critique 24

- Section 1.4: What fluence was used (“low electron dose”)? Does the upsampling algorithm imply the images presented in the manuscript are filtered? This should be stated in the captions if that is the case. If the raw data is available, it would be good to include either in the SI or in a data repository alongside any filtered/processed data.

Response to critique 24

We have partially answered to this critique in Critique 17.

We have made the following change:

Page S6 from:

To reduce e-beam damage, all images were recorded under low electron dose[...]

To:

To minimize the beam damage, the electron dose used was maintained under 700 e⁻/Å.

Critique 25

- Section 1.4: What are the inner and out collection angles for the ADF and ABF imaging conditions? What were the defocus values selected for image simulations and how were these determined (were a series of defocus values and thickness values explored and compared to determine the optimal match)? Is Cs the only aberration included? It is well known that, particularly low order aberrations, may drift over time (e.g. two-fold astigmatism and/or coma). How have these aberrations been handled in the simulations?

Response to critique 25

ADF and ABF imaging conditions were addressed in Critique 17.

For STEM imaging, we always aim for the best conditions (clearer image), thus for image simulations the defocus was set to 0. For image simulations we maintained optimum defocus conditions, as changing defocus only affected the clarity of the images (more or less blur), not like in TEM imaging. For thickness, indeed we simulated slices every 7.4 Å.

The software used includes, defocus (set to 0), astigmatism (we did not find significant differences within the range of ± 2 nm, the value we measured in the correction). Spherical aberration (below 1 micron) and comma (below ± 30 nm). Experimentally, we have corrected two-fold astigmatism and comma prior every image acquisition using the amorphous carbon.

Critique 26

- Section 1.4: What do the authors mean by “similarity to the experimental observation”? Does this mean that a judgment was made on the experimental images directly about the sample thickness or that the authors compared a thickness series of simulations to match the experimental data? If the latter, what thickness range was explored and how finely incremented were these simulations across different thicknesses?

Response to critique 26

We have updated the experimental section indicating the slice thickness that was 7.4 Å. From the several simulated images at different thickness, we selected the simulated image that was more similar to the experimental data.

Page S6 from:

Image simulations were performed using QSTEM software (C. T. Koch, Arizona State University, PhD thesis, 2002) based on the multislice simulation method. For that, two different supercells for the fcu and the reo phases were built with dimensions of 124.161 x 125.97 x 81.08 Å³ introducing same microscope parameters as experimentally used.

To:

Image simulations were performed using QSTEM software (C. T. Koch, Arizona State University, PhD thesis, 2002) based on the multislice simulation method (slice thickness 7.4 Å). For that, two different supercells for the *fcu* and the *reo* phases were built with dimensions of 124.161 x 125.97 x 81.08 Å³ introducing same microscope parameters as experimentally used.

Critique 27

- Figure S2: Fig S2c shows peaks that appear to be another phase for L/M 0.2-0.6. Have these been indexed and/or are they discussed in the text?

Response to critique 27

No, they have not. The focus of the paper is T = 120°C samples, the one with correlated defects:

From page 6:

Only low temperature samples (T ≤ 120 °C) showed appreciable superlattice reflections at low angular values that might be indicative of correlated defect domains in these solids, compared to the featureless diffraction patterns of the high temperature samples in the same region.

Critique 28

- Figure S9: The use of color bars covering different ranges of Mod/L ratios is visually confusing. It would be better to plot the colors consistently across all samples to support inter-comparison.

Response to critique 28

Figure S9 from:

To:

Critique 29

- Figure S19: It appears this is in fact the same particle shown in Fig. 6c but with some rotation applied. This information should be included as a clear statement in the caption here. The “ $T = 120\text{ }^{\circ}\text{C}$, $L/M = 0.4$ sample” could be read as being a particle from the same batch rather than the same particle. The rotation between the image here and Fig. 6c makes it less obvious that this is the identical particle shown across the two figures.

Response to critique 29

OK, the referee is right.

Figure S19 caption from:

Figure S19: Cs-corrected STEM-ADF analysis of $T = 120\text{ }^{\circ}\text{C}$, $L/M = 0.4$ sample displayed in Figure 6c

To:

Figure S19: Cs-corrected STEM-ADF analysis of the $T = 120\text{ }^{\circ}\text{C}$, $L/M = 0.4$ particle displayed in Figure 6c

Additional Changes

Page 10 from:

[...] (Figure SS10), that is an isoreticular analogue of the framework ZrFA built from $Zr_6O_4(OH)_4$ clusters and formate linkers.

To:

(Figure S10), that is an isoreticular analogue of the framework ZrFA built from $Zr_6O_4(OH)_4$ clusters and formate linkers.[Additional Reference]

[Additional Reference] Liang, W., Babarao, R., Murphy, M. J. & D'Alessandro, D. M. The first example of a zirconium-oxide based metal–organic framework constructed from monocarboxylate ligands. Dalton Trans. 44, 1516–1519 (2015).

Some of the author affiliations have changed.

We have modified the acknowledgement section to include:

ZD also acknowledges CñEM, School of Physical Science and Technology, ShanghaiTech University (#EM02161943)

Response to reviewers letter

To conclude, we would like to thank the Editor and the referees for their diligent work in evaluating our work. In line with their comments and suggestions, we have put forward an effort to improve the content, clarity and scholarship of the work that was originally submitted for publication. We hope that the result will be satisfactory and that the publication of the work will not be delayed any longer.

Yours sincerely,

Carlos Martí-Gastaldo,
Instituto de Ciencia Molecular (Universidad de Valencia)

REVIEWERS' COMMENTS

Reviewer #2 (Remarks to the Author):

The authors have made comprehensive and thorough revisions to the manuscript. I have noted a few points that I think would still improve the manuscript prior to publication, but I do not recommend any further peer review. I recommend publication in *Nature Communications* taking into account these minor revisions insofar as possible.

Reviewer 2, Critique 1

The authors write in response that "As the size of the crystals does not change with ligand concentration (Figure S15), net reduction of *fcu* fractions in the crystals will necessarily involve bigger *reo* fractions or domains." I am not sure I follow the logic here. Figure S15 presents SEM data of the particle size. Assuming these are single crystals given the faceting, the particle size may be interpreted as the total crystal size in this case within reason. However, the *reo* domains are known to be coherent. Could the authors clarify (in the article text) how they are able to distinguish between a larger number of small domains and a fixed number of domains of varying size to explain the observed increase in the *reo* fraction as a function of linker concentration? Although it may seem like a minor point, I think precision in distinguishing between fractions and domain size is important. An increase in domain size may explain increasing *reo* fraction, but a greater number of relatively small domains can contribute to a larger fraction as well. Maybe I am missing some detail (such as crystallite size information from powder XRD), but it may be that a revision to the text could clarify this point to readers having this same question in mind.

Reviewer 2, Critique 8

I would suggest that other readers may have a similar question about the choice of P1/P3 and P2/P3. Would a comment on this choice in the text offer improved clarity?

Reviewer 2, Critique 9

To clarify my previous comment, I might suggest the use of defect concentration or defect number in place of 'defectivity' as both of those alternatives would seem to be more interpretable than the term 'defectivity' itself.

Reviewer 2, Critique 15

I am very glad to see the addition of Fig. S18-S21 as this critically underpins the other presented STEM data.

Reviewer 2, Critique 17

I would restate my view that image processing applied, including image filtering, should be commented on in the caption (main text Figure 6). Additional details are appropriate in the SI, but the caption should make it clear how the data has been handled, particularly where denoising filters have been used.

I note also that there appears to be a typo in the reported dose, as the dose or fluence is typically reported as electrons per unit area (missing a power of two for the Ångstrom symbol).

Reviewer 2, Critique 24

How does the reported fluence (dose) compare with the estimated critical fluence (critical dose) for UiO-66? How was the critical fluence calculated? Usually, for STEM imaging this requires some estimate of the probe size together with the field of view and pixel size (probe sampling).

Are the authors planning to make the underlying data for the manuscript available (open data) on publication?

RESPONSE TO REVIEWERS' COMMENTS

We would like to express one last time our sincere thanks to the reviewer for their positive assessment and constructive criticism.

Reviewer 2

Comments

The authors have made comprehensive and thorough revisions to the manuscript. I have noted a few points that I think would still improve the manuscript prior to publication, but I do not recommend any further peer review. I recommend publication in Nature Communications taking into account these minor revisions insofar as possible.

Critique 1

The authors write in response that “As the size of the crystals does not change with ligand concentration (Figure S15), net reduction of **fcu** fractions in the crystals will necessarily involve bigger **reo** fractions or domains.” I am not sure I follow the logic here. Figure S15 presents SEM data of the particle size. Assuming these are single crystals given the faceting, the particle size may be interpreted as the total crystal size in this case within reason. However, the **reo** domains are known to be coherent. Could the authors clarify (in the article text) how they are able to distinguish between a larger number of small domains and a fixed number of domains of varying size to explain the observed increase in the **reo** fraction as a function of linker concentration? Although it may seem like a minor point, I think precision in distinguishing between fractions and domain size is important. An increase in domain size may explain

increasing **reo** fraction, but a greater number of relatively small domains can contribute to a larger fraction as well. Maybe I am missing some detail (such as crystallite size information from powder XRD), but it may be that a revision to the text could clarify this point to readers having this same question in mind.

Response to critique 1

We agree with the referee “an increase in domain size may explain increasing **reo** fraction, but a greater number of relatively small domains can contribute to a larger fraction as well”. In qualitative terms the size of the domains will be roughly proportional to the width of P2 (Scherrer equation: wider peaks smaller particles). In turn, the **reo** fraction as determined by Rietveld refinement determines the total fraction (number) of defective domains. In our case we see both a narrowing of the P1 peak along with an increase of the reo fraction, meaning that **reo** domains not only increase in number but also in size. However, as the shape factor is unknown to us, we preferred not to attempt a quantification of the crystallite size.

Critique 8

I would suggest that other readers may have a similar question about the choice of P1/P3 and P2/P3. Would a comment on this choice in the text offer improved clarity?

Response to critique 2

The following text have been added:

Additional Supplementary Note in Page 9:

We constated that I_{P2}/I_{P3} and I_{P1}/I_{P3} follow roughly the same trend with linker concentration. However, at very low **reo** fractions P1 tends to be wider than P2. Moreover, background contribution to the overall diffraction pattern is more intense at low angles making more difficult to measure I_{P1} than I_{P2} . That is why we decided to choose I_{P2}/I_{P3} .

Critique 9

To clarify my previous comment, I might suggest the use of defect concentration or defect number in place of ‘defectivity’ as both of those alternatives would seem to be more interpretable than the term ‘defectivity’ itself.

Response to critique 9

Following the referee suggestion:

“defectivity” has been changes to “defect concentration” wherever synonyms.

Critique 15

I am very glad to see the addition of Fig. S18-S21 as this critically underpins the other presented STEM data.

Response to critique 15

We are very pleased to read that this new version of the manuscript has met the referee's standards.

Critique 17.1

I would restate my view that image processing applied, including image filtering, should be commented on in the caption (main text Figure 6). Additional details are appropriate in the SI, but the caption should make it clear how the data has been handled, particularly where denoising filters have been used.

Response to critique 17.1

Following referee's suggestion, the following text have been added to the captions of Figure 6.

Figure 6 additional text:

All images have been denoised using HREM filters implemented in Digital Micrograph[Additional Reference]

[Additional Reference] Ishizuka, A.; Kimoto, K.; Ishizuka, K. Realtime Up-sampling Noise Filter: Paradigm Shift for Data Acquisition. *Microscopy and Microanalysis* 2020, 26, 1936–1938

Critique 17.2

I note also that there appears to be a typo in the reported dose, as the dose or fluence is typically reported as electrons per unit area (missing a power of two for the Ångstrom symbol).

Response to critique 17.2

We than the referee for pointing this out.

Page S6 from:

[...]dose used was maintained under $700 \text{ e}^{-}/\text{Å}$.

To:

[...]dose used was maintained under $700 \text{ e}^{-}/\text{Å}^2$.

Critique 24

How does the reported fluence (dose) compare with the estimated critical fluence (critical dose) for UiO-66? How was the critical fluence calculated? Usually, for STEM imaging this requires some estimate of the probe size together with the field of view and pixel size (probe sampling).

Response to critique 24

As reported in other works, the dose was calculated by knowing the current of the probe previously calibrated under the experimental conditions, then we also know the pixel size, therefore we can deduce the current per square area. And finally, we know the pixel time, time of the probe "sitting" at each pixel. For high-resolution imaging, pixel size is much smaller than the probe, thus the probe hits several pixels each time. This is not the most accurate determination for the electron dose, but should be a close approximation.

In what regards the critical dose, we have not quantitatively determined it for UiO-66 as this would require the analysis of several crystals to evaluate and measure the framework distortion depending on the electron dose. We believe that this parameter would also be affected by crystal thickness, location of particles (on the carbon support or on the hole); thus, we believed that measuring the critical dose for UiO-66, despite being very interesting and worthy for investigation, goes beyond the scope of this work.

Additional Question

Are the authors planning to make the underlying data for the manuscript available (open data) on publication?

The following section have been added:

Data Availability

The processed data associated to the manuscript Figures has been deposited in Zenodo under accession code 10.5281/zenodo.8282712. 'Crystallographic data for the structure reported in this Article have been deposited at the Cambridge Crystallographic Data Centre, under deposition numbers CCDC 2223895 (4c). Copies of the data can be obtained free of charge via <https://www.ccdc.cam.ac.uk/structures/>.

Additional datasets generated during the current study are available from the corresponding author on reasonable request.

Response to reviewers letter

To conclude, we would like to thank the Editor and the referees for their diligent work in evaluating our work. In line with their comments and suggestions, we have put forward an effort to improve the content, clarity and scholarship of the work that was originally submitted for publication. We hope that the result will be satisfactory and that the publication of the work will not be delayed any longer.

Yours sincerely,

Carlos Martí-Gastaldo,
Instituto de Ciencia Molecular (Universidad de Valencia)